# DISCOVER:
# Making Vision Networks Interpretable via Competition and Dissection

**Konstantinos Panousis**
Department of Electrical Eng., Computer Eng., and Informatics
Cyprus University of Technology
Limassol 3036, Cyprus
`k.panousis@cut.ac.cy`

**Sotirios Chatzis**
Department of Electrical Eng., Computer Eng., and Informatics
Cyprus University of Technology
Limassol 3036, Cyprus
`sotirios.chatzis@cut.ac.cy`

## Abstract

Modern deep networks are highly complex and their inferential outcome very hard to interpret. This is a serious obstacle to their transparent deployment in safety-critical or bias-aware applications. This work contributes to *post-hoc* interpretability, and specifically Network Dissection. Our goal is to present a framework that makes it easier to *discover* the individual functionality of each neuron in a network trained on a vision task; discovery is performed in terms of textual description generation. To achieve this objective, we leverage: (i) recent advances in multi-modal vision-text models and (ii) network layers founded upon the novel concept of stochastic local competition between linear units. In this setting, only a *small subset* of layer neurons are activated *for a given input*, leading to extremely high activation sparsity (as low as only $\approx 4\%$). Crucially, our proposed method infers (sparse) neuron activation patterns that enables the neurons to activate/specialize to inputs with specific characteristics, diversifying their individual functionality. This capacity of our method supercharges the potential of dissection processes: human understandable descriptions are generated only for the very few active neurons, thus facilitating the direct investigation of the network's decision process. As we experimentally show, our approach: (i) yields Vision Networks that retain or improve classification performance, and (ii) realizes a principled framework for text-based description and examination of the generated neuronal representations.

## 1 Introduction

In recent years, Deep Neural Networks (DNNs) have exhibited an overwhelming success in a variety of tasks and applications, achieving state-of-the-art results in Machine Vision, Automatic Speech Recognition, and NLP. This unprecedented success is however accompanied with a limited capacity to audit them in view of reliability standards. Indeed, due to the immense complexity of their architectures, DNNs are usually employed as off-the-shelf solutions for a variety of applications. However, their vast and highly accelerated adoption rate emphasizes the importance of being able to *explain* how DNNs function and *interpret* how predictions are made. This process will allow for

37th Conference on Neural Information Processing Systems (NeurIPS 2023).

uncovering their inherent biases and even for correcting the failures in their decision making process, facilitating a safe, trustworthy and robust real-world deployment in safety-critical applications.

Fortunately, this limitation has recently received a lot of attention in the Deep Learning (DL) research community, leading to the development of ways forward towards *Interpretable* Deep Networks. In this context, we can identify two major approaches: *ante-* and *post-hoc* methods. In the former, the main rationale is to build networks from scratch, that integrate intepretability components into the network itself; Concept Bottleneck Models (CBMs) are a characteristic case [13, 17, 19, 25]. In post-hoc approaches, backbone architectures are augmented with novel frameworks aiming to provide explanations of the prediction process, e.g. saliency maps [15, 27] and Network Dissection [2, 9]. Each approach exhibits its own advantages and drawbacks. For example, ante-hoc models usually suffer from significant performance drop, often requiring additional data to train. On the other hand, even though post-hoc models avoid the complexity of re-training, the results may lend themselves to subjective interpretations that depend on the method's assumptions.

This work takes a novel route: it aims at improving the efficacy of post-hoc methods by addressing some fundamental building blocks of Deep Networks. Specifically, we depart from the commonly used forms of non-linearities in Deep Networks and propose the use of competition-based activations imposed upon otherwise linear projection units; these form the so-called Stochastic Local Winner-Takes-All (LWTA) layers [22]. Our inspiration stems from biological arguments: it has been shown that brain neurons with similar functions aggregate together in blocks and locally compete for their activation; the winner gets to pass its response to other neurons outside the block while the rest are inhibited to silence [30, 1, 14]. Stochastic LWTA activations have recently been employed on DNNs with striking success, exhibiting: (i) significant compression capabilities [22], (ii) unique representation diversification ability [23], (iii) strong adversarial robustness against powerful adversarial attacks [24], (iv) the best ever reported performance in model-agnostic meta-learning (MAML), in terms of both accuracy and network footprint [11]. Other significant properties of LWTA-based networks include *noise suppression, specialization, automatic gain control* and *robustness to catastrophic forgetting*[29, 8, 3], while their prowess has also been shown in Transformer networks dealing with end-to-end translation of sign-language video into text [32].

In building interpretable networks, the proposed competition-based rationale comes with two important benefits: (i) a naturally arising activation sparsity: only the winner unit in each block passes its computed output to the next layer, while the rest pass zero values; and (ii) a data-driven pattern of neuron functionality, based on the probability of a unit being the active winner for a given input.

This mode of operation naturally lends itself to the *post-hoc* paradigm of unraveling and inspecting individual neuron functionality, which this work addresses. In this setting, recent works on automated methods aim to provide high quality descriptions of neurons by leveraging multi-modal models to match neuron functionality to text-based *unrestricted concepts*.

We leverage these ideas to address the interpretation limitations of conventional architectures, by introducing the **Dis**section of **Co**mpetitive **V**ision Netwo**r**ks (DISCOVER) approach, a novel framework towards interpretable Vision Networks. Our contributions can be summarized as follows:

- We construct, implement and train a variety of Competitive Vision Networks (CVNs) utilizing novel competition arguments, including both CNN and Transformer-based architectures. This is the first time in the literature that CVNs are trained on a large scale and their behavior is investigated with respect to both: (i) performance, and (ii) interpretation properties.

- We perform *post-hoc network dissection* using multimodal models to identify the underlying functionality and contribution of each individual neuron; to this end, we leverage the unique characteristics of competition between neurons and novel Network Dissection approaches.

- We examine the specialization properties of CVNs in the context of interpretability by investigating the generated per-example neuronal representations.

## 2 Background

**Image-Text Models.** Contrastive Language-Image Pre-training (CLIP) [26] constitutes one of the best-known methods for multimodal deep learning, combining visual representations with natural

language supervision. The main operating principle of CLIP relies on the usage of image-caption pairs in order to obtain representations. Specifically, CLIP comprises an image encoder, denoted by $E_T(\cdot)$ and a text encoder $E_I(\cdot)$ that are trained simultaneously on the grounds of a contrastive learning objective[28, 4]. Specifically, assuming a batch of $N$ (image, text) pairs, i.e. $\{(x_i, t_i)\}_{i=1}^N$, CLIP learns a *multi-modal* embedding space by maximizing the cosine similarity of the embeddings $I_i = E_I(x_i)$ and $T_i = E_T(t_i)$, between the $N$ correct pairs, while minimizing the cosine similarity between the other $N^2 - N$ combinations. During inference, we can use natural language supervision to perform zero-shot classification of any given image and labels. Thus, given a test image $x^{\text{test}}$ and some labels $\{t_j\}_{j=1}^M$, we can use the learned encoders and select the image-label pair that has the highest similarity of all the possible combinations.

**Network Dissection.**  Network Dissection constitutes one of the most popular approaches for understanding and describing the contribution of individual deep network units in the produced inferential outcomes. In [2], a method that yields descriptive outcomes in terms of *concepts* is presented. To this end, the authors introduce a newly developed densely-labeled dataset, named *Broden*; this contains pre-determined concepts, images and their associated pixel-level labels. The main principle is to use the Intersection over Union (IoU) score as a measure of relevance between each concept and neuron in the set. However, the proposed method requires the use of dense labels for training, while at the same time the concepts are restricted to the pre-defined label set.

This limitation inspired the development of MILAN [9], an automated method that addresses the notion of pre-defined concepts through the generation of unrestricted description via generative image-to-text models. Nevertheless, the requirement for human annotation is still present, similar to Network Dissection [2]. CLIP-Dissect [20] bypasses this restriction by leveraging multimodal image-text models, and specifically CLIP; this allows for decoupling the dependence between the concept set and the probing dataset. Thus, any network can be dissected, using any text corpus to form the concept set, any image probing dataset, and any appropriate similarity measure for matching concepts to neurons.

In this work, we exploit the flexibility of CLIP-Dissect and extend it in the context of Competitive Vision Networks. Differently than CLIP-Dissect, we mainly focus our analysis on the properties of Vision Transformer architectures with respect to neuron identification.

**Competitive Vision Networks.**  Recent works have explored a fundamentally different paradigm for the latent unit operation of DNNs, based on the concept of local competition among units (LWTA) [29]. In this setting, latent units compute their response and compete for their activation. The winner passes its (linear) response to the next layer, while the rest output zero values. In the following, we will be referring to these networks as Competitive Vision Networks (CVNs).

Let us consider an input $x \in \mathbb{R}^J$ presented to a hidden layer of a DNN comprising $K$ hidden units and a weight matrix $W \in \mathbb{R}^{J \times K}$. To compute a typical non-linear activation, each hidden unit in the layer performs an inner product computation with its respective weights $w_k \in \mathbb{R}^J$ yielding the intermediate response $h_k = w_k^T x \in \mathbb{R}$; this usually passes through a non-linear activation $\sigma(\cdot)$, such that the final layer output reads: $y = [y_1, \ldots, y_K]$, where $y_k = \sigma(h_k), \forall k$.

In a corresponding LWTA-based layer, $U$ singular units are grouped together forming the so-called LWTA blocks. Hereinafter, we denote by $B$ the number of LWTA blocks in the layer, and with $U$ the number of *linear competing units* therein. The aggregation operation is manifested via the definition of the layer's weights as a three-dimensional matrix $W \in \mathbb{R}^{J \times B \times U}$; this structural modification implies that each input dimension of $x$ is now presented to each block $b$ and each unit $u$ therein; in turn, all units compute their response via the standard inner-product computation $h_{b,u} = w_{b,u}^T x \in \mathbb{R}, \forall b, u$ and competition takes place among them. The fundamental principle of competition is that out of the $U$ competing units, *only one can be the winner*. This unit gets to convey its activation outside the block, i.e. the next layer, while the rest output zero values. This process is instantiated via an appropriate competition function that encodes the outcome of the competition in each block.

Contrary to the conventional definition, where the layer's output arises as the concatenation of the individual response of each unit, in the LWTA-based framework, the final output $y^{B \cdot U}$ is now constructed from $B$ sub-vectors, one for each LWTA block that contains a *single non-zero entry*; this corresponds to the response of the winner unit in said block. Thus, an inherent property of the LWTA mechanism is the naturally emerging *sparse representations*. Indeed, considering a fixed number of

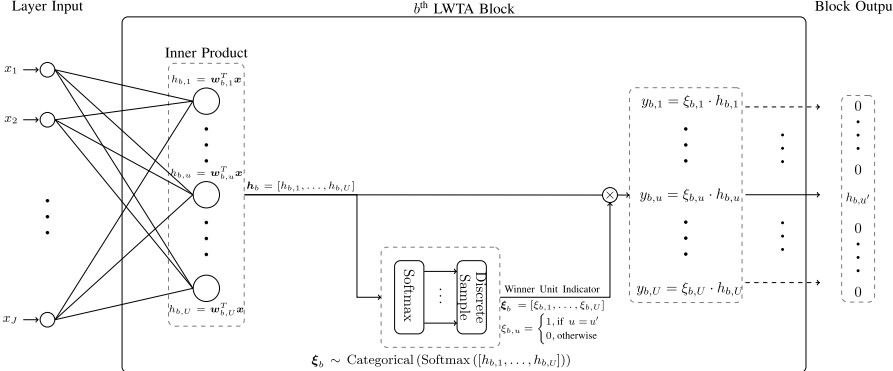

Figure 1: Detailed bisection of the $b^{\text{th}}$ Stochastic LWTA block. Presented with an input $\boldsymbol{x} \in \mathbb{R}^J$, each unit $u$ computes its activation $h_{b,u}$ via different weights $\boldsymbol{w}_{b,u} \in \mathbb{R}^J$, i.e., $h_{b,u} = \boldsymbol{w}_{b,u}^T \boldsymbol{x}$. The linear responses are concatenated, s.t., $\boldsymbol{h}_b = [h_{b,1}, \dots, h_{b,U}]$, and transformed into probabilities via the softmax operation. Then, a Discrete sample $\boldsymbol{\xi}_b = [\xi_{b,1}, \dots, \xi_{b,U}]$ is drawn, which is a one-hot vector with a single non-zero entry at position $u'$, denoting the winner unit in the block; unit $u'$, passes its linear response to the next layer, while the rest pass zero values. Image from [23].

units per layer, we can observe that the higher the number of competitors $U$, the sparser the layer output: when $U = 2$, only $50\%$ of the units will be active per example, when $U = 4$ only $25\%$ and so on. This gives rise to a sparse activation mode of operation that has been empirically shown to endow deep architectures with significant properties including strong adversarial robustness [24] and substantial representation diversification [23]. One design choice that we need to address concerns the nature of the competition: *deterministic* or *stochastic*. The former is the most typical approach in the LWTA literature; in this case, the unit with the highest activation is deemed the winner each time. The latter is founded upon novel stochastic arguments proposed in [22]. This formulation has been shown to consistently outperform its deterministic counterpart [21, 24, 32, 11]. Thus, we adopt stochastic competition in our approach.

The stochastic variant of the competition function entails the introduction of an appropriate set of discrete latent variables for each LWTA block $b$, $\boldsymbol{\xi}_b \in \text{one\_hot}(U)$, that encode the (stochastic) outcome of the competition therein. The latent indicators constitute one-hot vectors with $U$ components; that is, vectors containing a *single non-zero* entry at the *index position* corresponding to the winner unit in the block. The output $\boldsymbol{y}$ of a stochastic LWTA layer's $(b, u)^{th}$ component $y_{b,u}$ yields:

$$y_{b,u} = \xi_{b,u} \sum_{j=1}^{J} w_{j,b,u} \cdot x_j \in \mathbb{R} \tag{1}$$

We postulate that these latent indicators $\boldsymbol{\xi}_b, \forall b$, are drawn from a Categorical distribution driven from the intermediate linear inner-product computations that each unit performs. Evidently, the higher the response of a particular unit in a block (relative to the others), the higher its probability of being the winner; however, the final decision remains stochastic. We can formulate this rationale as:

$$q(\boldsymbol{\xi}_b) = \text{Categorical}\left(\boldsymbol{\xi}_b \middle| \Pi_b(\boldsymbol{x})\right), \ \Pi_b(\boldsymbol{x}) = \text{Softmax}\left(\sum_{j=1}^{J} [w_{j,b,u}]_{u=1}^{U} \cdot x_j\right), \quad \forall b \tag{2}$$

where $\Pi_b(\boldsymbol{x})$ is a vector comprising the *activation probability* of each neuron in the block, and $[w_{j,b,u}]_{u=1}^{U}$ denotes the vector concatenation of the set $\{w_{j,b,u}\}_{u=1}^{U}$. Each unit in each block computes its intermediate linear inner-product; these are then passed to a softmax transformation and used to draw samples from a categorical distribution. The resulting discrete vector contains a single non-zero entry denoting the winner of the block. The intermediate computation of the winner unit passes out as its activation; the rest units pass zero-valued activations. A graphical illustration of the stochastic LWTA structure is provided in Fig. 1.

In the context of attention-based architectures, the encoder of Vision Transformers (ViTs) [6] comprises alternating layers of Multi-head Self-Attention (MSA), MLP blocks, Layer Normalization

(LN) and residual connections. Each MLP block comprises two layers with a GELU non-linearity in between. Within the CVN framework, we replace GELU layers with stochastic LWTA layers. This modeling decision is motivated by recent works that have shown that transformer MLPs encode knowledge attributes [5, 7, 18]. We posit that this formulation facilitates neuron identification and interpretation via analysis of the arising competition patterns; we explore its potency in the following sections.

Further, convolutional operations also constitute an integral part of various architectures. To account for this fact, and for completeness, we also consider the convolutional variant of the stochastic LWTA layer proposed in [22]. Due to space constraints, and since we mainly focus our analysis on attention-based architectures, we provide the formulation in the Supplemental Material.

## 3 DISCOVER

### 3.1 Training and Inference Algorithms for CVNs

CVNs comprise additional auxiliary variables denoting the winner in each LWTA block; thus, we need to devise an appropriate training regime that takes into consideration the stochastic nature of said variables. To this end, we turn to Stochastic Gradient Variational Bayes (SGVB) [12], and construct an Evidence Lower Bound (ELBO) objective. Assuming a dataset $\mathcal{D} = \{\boldsymbol{X}_i, y_i\}_{i=1}^N$, and denoting as $f(\boldsymbol{X}_i; \hat{\boldsymbol{\xi}})$ the cross-entropy between the target labels $y_i$ and the target probabilities emerging from a CVN, the objective reads:

$$\mathcal{L}^{\mathrm{CVN}} = -\sum_{\boldsymbol{X}_i, y_i \in \mathcal{D}} \mathrm{CE}\Big(y_i, f(\boldsymbol{X}_i; \hat{\boldsymbol{\xi}})\Big) - \mathrm{KL}[q(\boldsymbol{\xi})||p(\boldsymbol{\xi})] \tag{3}$$

where $\hat{\boldsymbol{\xi}}$ denotes *samples from the (posterior) distributions* of all latent variables, $\boldsymbol{\xi}$, in all layers, and $\mathrm{KL}[q(\boldsymbol{\xi})||p(\boldsymbol{\xi})]$ denotes the Kullback-Leibler divergence between the posterior and the prior. We assume a symmetric Categorical prior for the latent variable indicators $\boldsymbol{\xi}$; hence, $p(\boldsymbol{\xi}_b) = \mathrm{Categorical}(1/U), \forall b$. For all the computations we draw just one sample, but we consider it as a differentiable expression; this allows for very low variance gradients during training, which ensures convergence (reparameterization trick). Since the Categorical distribution is not amenable to differentiation, we enable reparameterization by resorting to a continuous relaxation, namely Gumbel-Softmax (GS) [16, 10]. In contrast to previous works [22, 23], we use the Straight-Through variant of GS, which we have seen to offer better convergence properties. We present the exact sampling procedure in the Supplemental Material. During inference, we directly draw samples from the trained posteriors and determine the winning units in each layer along with their responses (Eq. 2).

### 3.2 Dissection

For dissecting the considered CVNs, we draw inspiration from CLIP-Dissect [20]. We employ a CLIP model to identify the functionality of each neuron in a stochastic LWTA block in terms of a given *concept* set; this will be matched with a textual description. There are three key components that define the approach: (i) a concept set $\mathcal{S}$, where $|\mathcal{S}| = M$, (ii) a dataset that is used as a probe to identify the individual functionality of neurons denoted by $\mathcal{D}_{\mathrm{probe}}$, where $|\mathcal{D}_{probe}| = N$, and (iii) the network to be investigated, denoted by $f(x)$. The main principle is to match neuronal activations with concepts using: (i) the *activation matrix* $\boldsymbol{P}$ that measures the similarity between images and concepts, and (ii) a summary of the individual neuronal activations of the network being probed to the given probe dataset.

**Concept Activation Matrix.** We use the image and text encoder of the CLIP model, denoted by $E_I$ and $E_T$ respectively, and compute the respective embeddings $I_i$ and $T_i$ for each image $x_i$ and concept $t_i$. This results in a concept activation matrix $\boldsymbol{P} \in \mathbb{R}^{N \times M}$, where each entry $(i, j)$ denotes the cosine similarity between image $x_i$ and concept $t_j$ in the CLIP embedding space.

**Record Activations for each Competitive Neuron.** Given a neuron $k$ and for each image in the probe dataset $\{\boldsymbol{X}_n\}_{n=1}^N$, CLIP-Dissect computes the neuron's non-linear responses $\{A_k(\boldsymbol{X}_n)\}_{n=1}^N$, and records them into a single vector $\boldsymbol{q}_k \in \mathbb{R}^N$, s.t., $\boldsymbol{q}_k = [A_k(\boldsymbol{X}_1), \ldots, A_k(\boldsymbol{X}_N)]$. Each *activation vector* $\boldsymbol{q}_k$ essentially *encodes* the *activation* of neuron $k$ across the whole probing dataset.

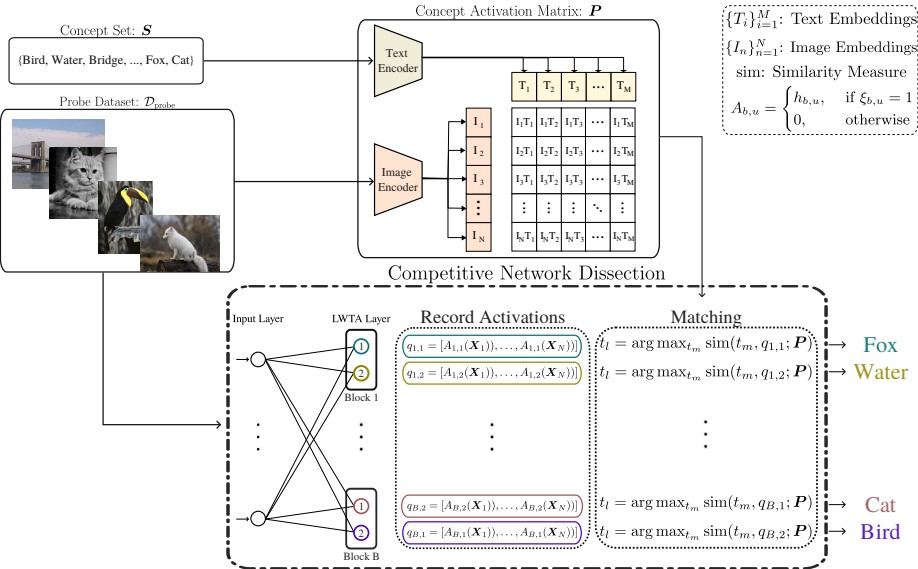

Figure 2: The DISCOVER framework. Given a probe dataset $\mathcal{D}_{\text{probe}} : \{\boldsymbol{X}_n\}_{n=1}^N$, a concept set $\boldsymbol{S}, |\boldsymbol{S}| = M$, and a layer to be dissected: compute the concept activation matrix $\boldsymbol{P} \in \mathbb{R}^{N \times M}$ using CLIP; then, for each LWTA block $b$ and each neuron $u$ therein, compute the responses $A_{b,u}(\boldsymbol{X}_n), \forall n$, s.t. $q_{b,u} = [A_{b,u}(\boldsymbol{X}_1), \dots, A_{b,u}(\boldsymbol{X}_N)]$. Each neuron $b, u$ is then matched to the concept $t_l$ whose activation vector $\boldsymbol{P}_{:,l}$ has the highest *similarity* (sim) with the neuron's activation vector $\boldsymbol{q}_{b,u}$.

Subsequently, it can be exploited as a data-wide neuron representation to match neurons to textual descriptions as we describe next. When the response $A_k(\boldsymbol{X}_n)$ is not a scalar, a *summary* function that maps it to a real number is employed. For inner-product neurons, the response is already a real-number. For convolutional kernels, we consider a mean over the spatial dimensions.

**Our approach.**   Building on these outcomes, CVNs adopt a novel regard toward interpretation, which leverages the sparse nature of the representations obtained from stochastic LWTA layers; here, we focus on attention-based architectures. In this setting, the MLP blocks contain patch-specific information relating to the tokens of each image; in principle, we could introduce a summary function to aggregate the patch-wise information and output a real-number. However, we have access to another representation: the *feature embedding* pertaining to the *class token*; this comprises a set of units dedicated to capturing both global and local contextual information in each layer.

Within the framework of CVNs, this is treated in a local competition fashion: CVN units on each MLP are grouped together in blocks of competitors and compete for the output of the block. Contrary to the singular recording of activations in the base formulation for each neuron $k$, $\{A_k(\boldsymbol{X}_n)\}_{n=1}^N$, the activations are computed for each LWTA block $b$ and unit $u$ therein, yielding $\{A_{b,u}(\boldsymbol{X}_n)\}_{n=1}^N$. These activations can be negative, positive or exactly zero, depending not only on the linear computation of the unit in consideration, but also the responses of the remaining units in its LWTA block, following the form of Eq.1. We concatenate them to construct a summary for each competitive unit, such that $\boldsymbol{q}_{b,u} = [A_{b,u}(\boldsymbol{X}_1), \dots, A_{b,u}(\boldsymbol{X}_N)]$.

Due to the nature of the proposed stochastic local competition, the emerging summary vectors will be *sparse*; only the winner of each LWTA block will retain its linear computation and pass it as a non-zero activation for each example $\boldsymbol{X}_n$; the rest units in the block pass zero activations. This mode of operation not only encourages specialization, but can also diversify individual neuron identification; we explore its potency in the next section.

**Matching Neurons to Concepts.**   Having recorded the activations vectors for each LWTA block $b$ and each neuron $u$ therein, $\boldsymbol{q}_{b,u} \in \mathbb{R}^N$, we aim to discover the most *similar* concept $\{t_m\}_{m=1}^M$ to describe each neuron. To achieve this matching, a similarity function $\text{sim}(t_m, \boldsymbol{q}_{b,u}; \boldsymbol{P})$ is defined; this is used to quantify the relation (in terms of similarity) between the neuron's activation vector

$\boldsymbol{q}_{b,u} \in \mathbb{R}^N$ and the concept's $m$ activation vector, $\boldsymbol{P}_{:,m} \in \mathbb{R}^N$. For example, considering the *cosine similarity*, we compute:

$$\text{sim}(t_m, \boldsymbol{q}_{b,u}; \boldsymbol{P}) \propto \boldsymbol{P}_{:,m} \boldsymbol{q}_{b,u}^T \tag{4}$$

Thus, for assigning a concept to neuron $b, u$, we compute its similarity to each concept in the set and select the one that exhibits the highest value, s.t., $t_l = \arg\max_m \text{sim}(t_m, \boldsymbol{q}_{b,u}; \boldsymbol{P})$. Characteristic similarity functions include, Rank Reorder, WPMI and SoftWPMI [20].

At this point, it is important to highlight a principal benefit of CVNs: After neuron identification, i.e., matching neurons to concepts, and since only a *small subset* of neurons is active for each example, it becomes practically tractable to perform a per-example analysis on that particular small subset of "winner" (active) neurons; this greatly facilitates practical concept interrogation.

## 4  Experimental Analysis

For evaluating and dissecting the proposed CVNs, we train two sets of models: (i) Transfomer-based, and (ii) Convolutional architectures. We consider stochastic LWTA layers with different numbers of competitors, ranging from $U = 2$ to $U = 24$. In every architecture, we retain the total number of parameters of the conventional model by splitting a layer comprising $K$ singular neurons to $B$ blocks of $U$ competing neurons, such that $B \cdot U = K$. This choice facilitates a fair -sizewise- comparison between an original network and its CVN counterpart. The number of competitors has a direct effect on the per example neuron activation in the respective layers. For example, when $U = 2$, only $50\%$ of neurons are activated for a given input, when $U = 8$ only $12.5\%$ and so on.

For the Transformer architecture, we select the DeiT model, specifically DeiT-Tiny (DeiT-T, 5M parameters) and DeiT-Small (DeiT-S, 22M parameters), which we train from scratch on ImageNet-1k. For the convolutional paradigm, we chose the same network as in CLIP-Dissect [20] for comparability; that is ResNet-18 trained on Places365. In all cases, we follow the original training schemes concerning the hyperparameters, and remove the excessive augmentations of DeiT, i.e., DropPath, Color Jittering, Random Erase, CutMix and MixUp; we have found that these do not improve accuracy of CVNs. For completeness, all hyperparameter settings for each model are provided in the Supplementary Material. All models were trained on a single NVIDIA A6000 GPU. Our code implementation is available at: `https://github.com/konpanousis/DISCOVER`.

**Accuracy.**  We begin our evaluation with classification results pertaining to CVNs and how block size (number of competitors in each block) affects performance. This is essential, since CVNs have not been explored in the literature, especially when dealing with larger datasets and architectures, and no baselines exist for the considered setup. However, it is highly important to note that this work is not focused on achieving state-of-the-art performance for vision tasks using CVNs. Instead, our focus is on providing a fundamental component towards a novel Network Dissection paradigm. Thus, we focus on Network Dissection efficacy, as opposed to optimally tuning hyperparameters and augmentation to increase accuracy by few points.

The obtained comparative results are presented in Table 1. Therein, we observe that despite the fact that we did not aim for performance improvements, CVNs exhibit near or even superior performance compared to their conventional counterparts. This finding persists even when using a large number of competitors, which directly translates to high activation sparsity. Let us consider for example DeiT-T/16; this model comprises LWTA blocks with 16 competing units each, leading to $1/16 = 6.25\%$ active (winning) neurons in each layer. We observe that this network exhibits only a $1\%$ drop in prediction accuracy (again without any tuning). On the other hand, when using a smaller number of competitors, e.g., DeiT-T/2 or DeiT-T/8, we obtain better and on par performance to the conventional architecture, respectively. Also note that, even though there is a minor computational overhead during training (up to $10\%$), the computational costs of inference are comparable. Thus, accuracy-wise, CVNs provide a realistic alternative to conventional Vision Networks that can even yield accuracy improvements when trained with the same (potentially suboptimal) setup.

**Quantitative Analysis.**  We follow the novel proposal of [20] to compare our CVN-based approach to conventional non-competitive networks. In this context, we can compare the generated neuron labels, for a specific layer of a network, with the ground truth descriptions, i.e., the class labels. Specifically, we measure the cosine similarity in a text embedding space between the ground truth

Table 1: Classification results for ImageNet and Places365. */None* denotes a non-competitive architecture, e.g., using ReLU/GELU, while numbers denote the number of competitors $U$.
{$\star$/ $*$}Locally reproduced results due to unavailability of pretrained models/resource limitations.

| **ImageNet-1k** | | |
|---|---|---|
| Model/Competitors | Accuracy + Std ($\%$) | Active Neurons Proportion |
| DeiT-T/None | $72.2 \pm$ N/A | Data Specific |
| DeiT-T/2 | $\mathbf{72.5} \pm 0.10$ | 0.500 |
| DeiT-T/8 | $71.7 \pm 0.15$ | 0.125 |
| DeiT-T/16 | $71.1 \pm 0.25$ | 0.062 |
| DeiT-T/24 | $70.5 \pm 0.45$ | 0.041 |
| DeiT-S/None | $77.0 \pm$ N/A$^*$ | Data Specific |
| DeiT-S/2 | $\mathbf{77.3} \pm 0.30$ | 0.500 |
| DeiT-S/12 | $77.0 \pm 0.25$ | 0.083 |
| DeiT-S/16 | $76.7 \pm 0.50$ | 0.062 |
| **Places365** | | |
| ResNet-18/None | $52.25 \pm$ N/A$^\star$ | Data Specific |
| ResNet-18/2 | $\mathbf{53.9} \pm 0.20$ | 0.500 |
| ResNet-18/4 | $51.0 \pm 0.50$ | 0.250 |
| ResNet-18/8 | $49.5 \pm 0.75$ | 0.125 |

class name and the description of the neuron. We use two different text encoders: (i) CLIP ViT-B/16, denoted as CLIP cos and (ii) all-mpnet-base-v2, denoted as mpnet cos. Here, we focus on the last layer of the MLP block of the DeiT-T model. The obtained comparative results are depicted in Table 2. Therein, we observe that the CVN *consistently* outperforms its non-competitive counterpart when using both the best-performing softWPMI and the Cubed Cosine Similarity function proposed in [20].

Table 2: Cosine similarity between the last layer neuron descriptions and ground truth labels on DeiT-T/None and DeiT-T/16 trained on ImageNet. For the former we use softWPMI for identification, while for CVNs, softWPMI and the cubed cosine similairity measure [20]. We use two text encoders: (i) CLIP ViT-B/16 and (ii) all-mpnet-base-v2, denoted as mpnet cos.

| $\mathcal{D}_{\text{probe}}$ | Concept Set $\mathcal{S}$ | Vision Network | | | | | |
|---|---|---|---|---|---|---|---|
| | | DeiT-T/None (SoftWMPI) | | **DeiT-T/16** (SoftWPMI) | | **DeiT-T/16** (CosSimCubed) | |
| | | CLIP cos | mpnet cos | CLIP cos | mpnet cos | CLIP cos | mpnet cos |
| ImageNet Val | Broden | 0.6250 | 0.1800 | 0.6460 | 0.1829 | **0.6543** | **0.1918** |
| —"— | 3k | 0.6226 | 0.1688 | 0.6650 | **0.1895** | **0.6699** | 0.1842 |
| —"— | 10k | 0.6187 | 0.1715 | 0.6553 | **0.1878** | **0.6616** | 0.1831 |
| —"— | 20k | 0.6128 | 0.1758 | 0.6455 | **0.1929** | **0.6519** | 0.1899 |
| —"— | ImageNet | 0.5850 | 0.1782 | **0.5854** | **0.1842** | 0.5840 | 0.1835 |
| CIFAR100 Train | 20k | 0.6392 | 0.1765 | **0.6533** | **0.1969** | 0.6530 | 0.1876 |
| Broden | 20k | 0.6338 | 0.1780 | 0.6470 | 0.1815 | **0.6606** | 0.1546 |
| ImageNet Val & Broden | 20k | 0.6299 | 0.1821 | 0.6333 | 0.1876 | **0.6729** | 0.1433 |

To further assess the quality of the obtained neuronal descriptions, we turn to the *neuron identification accuracy* metric proposed in [20]. In this context, and in situations where the considered concept set contains *exact class labels*, we compute the percentage of neurons that have been assigned the *exact correct* label. Evidently, this metric pertains to the classification layer of a considered network. We consider four datasets: *CIFAR-100 Train*, *Broden*, *ImageNet Val* and *ImageNet Val & Broden*; we use the ImageNet classes as the concept set $\mathcal{S}$. For matching neurons to concepts, we use Soft-WPMI, since it outperformed all other similarity functions. Further results with the other similarity metrics are provided in the Supplemental Material.

The comparative results for all conventional and competitive configurations are presented in Table 3. We observe that *on average*, CVNs consistently outperform their conventional ReLU/GELU based counterparts, with up to $2\%$ improvement. It is striking, that DeiT-S/12 improves accuracy on *CIFAR-100 Train* by a staggering $10\%$ compared to the conventional ResNet-50 that has the same number of parameters; at the same time, DeiT-T/8, also provided a significant improvement of $\approx 7\%$.

Table 3: Accuracy as the percentage of neurons assigned the correct label, i.e., class name. Concept Set $\mathcal{S}$: ImageNet, Similarity: SoftWPMI. ResNet-50 results from [20].

| $\mathcal{D}_{\text{probe}}$ | Vision Network | | | | | |
|---|---|---|---|---|---|---|
| | ResNet-50/None | DeiT-T/None | DeiT-T/8 | DeiT-T/16 | DeiT-S/2 | DeiT-S/12 |
| CIFAR-100 Train | 46.20% | 53.00% | 53.10% | 50.80% | 55.20% | **56.10**% |
| Broden | **70.50**% | 68.80% | 67.40% | 67.40% | 68.30% | 68.10% |
| ImageNet Val | 95.00% | 95.00% | **96.00**% | 95.30% | 95.60% | 95.60% |
| ImageNet Val & Broden | 95.40% | 95.20% | **95.70**% | 95.50% | 95.00% | 94.80% |
| Average | 76.78% | 78.00% | 78.05% | 78.30% | **78.70**% | 78.65% |

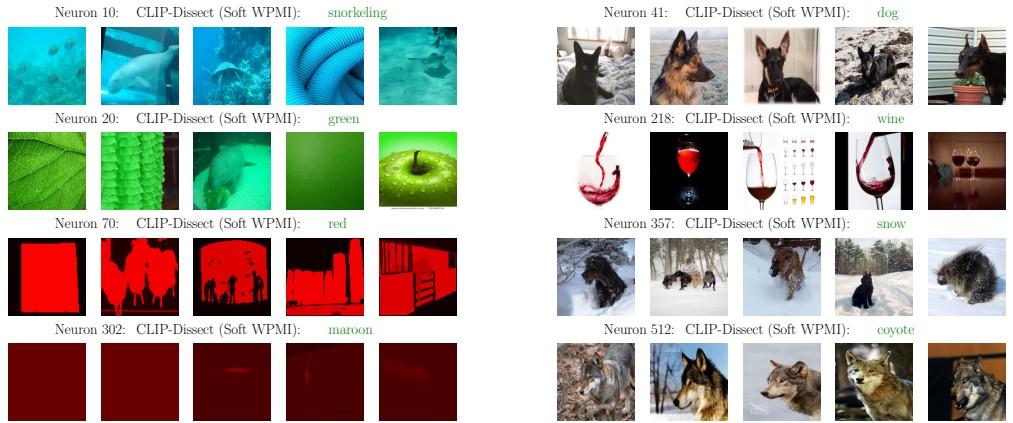

Figure 3: Neuron Identification for the first and last DeiT-T/16 MLP blocks using: SoftWPMI, $\mathcal{D}_{\text{probe}}$: ImageNet & Broden, $\mathcal{S}$: $20K$ most common English words [20].

**Qualitative Analysis.** We now turn to a qualitative analysis of the obtained neuronal representation. In this context, we visualize several neuron identification results in terms of generated descriptions in Fig. 3. To this end, we select random neurons of the first and last MLP blocks of DeiT-T/16 and use a combination of the ImageNet and Broden datasets as a probe; the concept set $\mathcal{S}$ comprises the $20,000$ most common English words similar to [20]. Since we do not use the same pretrained backbone, and the considered CVNs were trained from scratch, there isn't an exact matching between the neurons presented therein and the neurons of our networks, enumeration-wise. Thus, we can not make direct comparisons pertaining to the exact same neuron on an illustration basis. Nevertheless, we observe that in the context of CVNs we obtain highly accurate descriptions of the functionality of the networks' neurons. We observe that the randomly selected neurons are activated by semantically similar inputs that may contain only a part of the matching concept.

A principal property of CVNs is the per-example sparsity that naturally arises due to the competition mechanism; this can facilitate specialization and accelerate the process of examining the active concepts for each test point, greatly enhancing the interpretability of the network. Indeed, we perform this analysis for the last MLP block of the GELU-based DeiT-T/None, and its DeiT-T/8 and DeiT-T/16 competitive counterparts. This block comprises 768 neurons; for each example, DeiT-T/8 will activate only $768/8 = 96$ neurons and DeiT-T/16 only $48$ neurons. The number of activated neurons for the conventional DeiT-T/None will vary according to the probing dataset. For *ImageNet Val*, we found that on average, $98\%$ of neurons are activated for each example when using DeiT-T/None.

Thus, in Table 4, and for each architecture, we present the 7 most relevant concepts tied to neurons with the highest magnitude of activation to the considered example, along with 6 concepts tied to neurons with the lowest magnitude. Therein, we observe that the most contributing concepts in the conventional setting contain descriptions that are highly irrelevant or even contradicting to the presented input as "red/orange beak & legs", "black coat/tan markings" and "reddish brown fur"; at the same time $757/768$ neurons are active for this example, rendering the concept investigation a strenuous process. In stark contrast, the considered CVNs exhibit substantially more relevant concepts, while allowing for a straightforward examination of the emerging per-example concepts. Similar qualitative visualizations for different settings can be found in the Supplemental Material.

Table 4: Concepts tied to activated neurons for a random image from the ImageNet validation set for both DeiT-T/None and DeiT-T/16. For the former, $757/768 = 98.56\%$ neurons are activated leading to an arduous interpretation process; for DeiT-T/16 only $48/768 = 6.25\%$ of neurons are active.

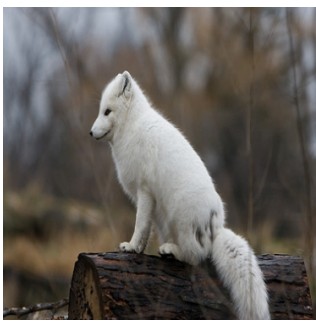

| DeiT-T/None | | DeiT-T/8 | | DeiT-T/16 | |
|---|---|---|---|---|---|
| Concepts | Act | Concepts | Act | Act | Concepts |
| white fur under tail | +3.06 | white fur under tail | +10.5 | fluffy/white appearance | +11.6 |
| red/orange beak & legs | +2.53 | a cucumber plant | +9.32 | large/fluffy white dog | +9.89 |
| white plumage | +2.19 | a fox | +8.90 | large/fluffy white dog | +7.71 |
| black coat/tan markings | +2.14 | pale/gray or cream fur | +8.36 | egret | +7.37 |
| large/fluffy white dog | +1.46 | white plumage | +7.33 | white/cream coloured coat | +7.07 |
| reddish/brown fur | +1.03 | white/gray plumage | +7.27 | blue/gray body | +6.87 |
| shaggy/gray coat | +0.71 | cabbage-like leaves | +7.04 | white plumage | +5.96 |
| a fox | −0.17 | small/black dog | +1.57 | a cub | +1.29 |
| large/fluffy white dog | −0.17 | shaggy/black-brown coat | +0.56 | wolf-like appearance | +0.96 |
| chicken | −0.17 | product displays | +0.46 | bandage | +0.68 |
| klin | −0.17 | a large/elegant dog | +0.34 | egg-laying mammal | +0.37 |
| sharp talons/beak | −0.17 | a bus driver | +0.18 | large/fluffy white dog | +0.35 |
| camel rider | −0.17 | an insect | +0.08 | black/tan coat | +0.08 |

# 5 Limitations & Conclusions

**Limitations.** Considering the use of the CLIP-Dissect approach for identifying neuron functionality in CVNs, this comes with its respective limitations, such as disregarding the spatial information; this, could allow for identifying lower level concepts and patterns. However, the structure of CVNs could potentially accommodate such a functionality via block (high) level and unit (low) level descriptions; we leave these explorations for future work. Finally, even though incorporating the competition mechanism to Vision Networks is a quite straightforward task, they aren't still implemented in an optimized way in popular frameworks, due to their limited adoption.

**Conclusions.** In this work, we introduced DISCOVER, a novel framework towards Interpretable vision architectures through Competitive Vision Networks and Network Dissection. For the first time in the literature, we trained LWTA networks using sufficiently large datasets and architectures. Despite the fact that we did not perform any hyperparameter tuning, the resulting networks yielded on par or even improved performance, even when using up to $4\%$ of active neurons per example. Our qualitative and quantitative results vouch for the efficacy and interpretation capabilities of our approach. We yielded highly interpretable neuron functionality, while using a small and explorable subset of all the available neurons per datapoint.

# Acknowledgements

This work has received funding from the European Union's Horizon 2020 research and innovation program under grant agreement No 872139, project aiD.

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

# A  Supplemental Material

## A.1  Experimental Details

Integrating the competition mechanism in modern architectures constitutes a pretty straightforward task. One just needs to replace the existing calls to the conventional non-linear activations with a call to a module/function implementing the LWTA rationale. This can be performed in-place without any other changes necessary. Thus, in the accompanied code, we can directly change the definitions of the Transformer or ResNet based models and train the model in the same manner as in the conventional Vision Networks. In this context, for the DeiT architectures, we alter the definitions in the *timm* library, while for ResNet-based architectures, we slightly alter the example implementation found in the official Pytorch repository[1]. For dissecting CVNs, we use the official repository of CLIP-Dissect [20]. We describe all changes in the respective README file. Our code and trained models will be public after publication.

**Transformers.** For training the CVN counterparts of the DeiT-T and DeiT-S models, we use the official implementation.[2] We train both architectures from scratch using ImageNet-1k for 300 epochs with the default parameters found therein. Specifically, we use a 5-epoch warm-up period, starting with an initial learning rate of $10^{-6}$, following a cosine annealing schedule up to $5 \cdot 10^{-4}$. We use the same AdamW optimizer and changed the used weight decay from $0.05$ to $0.02$ since we found that it hurt performance. We re-run the conventional GELU-based architectures with this selection and observed no change in the obtained accuracy. In contrast, we turned off the excessive augmentation setup, since it hurt the performance of CVNs. In this context, and in line with the initial ablation study presented in [31], we found that performance deteriorates when augmentations are removed in GELU-based networks. For training, we use a single sample for the Monte Carlo sampling estimation for the Evidence Lower Bound loss, described in the main text. For this, we turn to the continuous relaxation of the categorical distribution that allows for reparameterized samples and low variance estimation, as we describe in the next section. During inference, we draw 4 samples, average the logits in the Bayesian Averaging sense; we then compute the predicted loss and accuracy. We did not observe any improvement when drawing more samples.

**ResNet.** For training the ResNet-18 model, we used the ResNet ImageNet example PyTorch script and adapted the data loading for Places 365. We train the model for 90 epochs, using SGD with an initial learning rate of $0.1$ that is reduced by a factor of $0.1$ every 30 epochs, a weight decay of $10^{-4}$ and $0.9$ momentum. The batch size was set to 256. For the Gumbel-Softmax trick, we set the temperature to $0.67$ and used the Straight-Through estimator. During training, we only draw one sample for the reparameterization trick, while during inference we draw 4 samples from the trained posterior. We trained both conventional and CVN architectures since the official pretrained models were not available online. For both methods, we used the standard RandomResizedCrop and RandomHorizontal Flip augmentations.

## A.2  The Gumbel-Softmax Trick

In our work, we perform Monte-Carlo sampling using a single reparameterized sample for each of the corresponding latent variables. These are obtained via the reparameterization trick of the continuous relaxation of the Categorical distribution [16, 10] as described next. We focus on the reparameterization trick for the dense case; the convolutional case is analogous.

As a reminder, the latent indicators $\boldsymbol{\xi}_b, \forall b$, are drawn from a Categorical distribution driven from the intermediate linear inner-product computations that each unit performs.of $q(\boldsymbol{\xi})$ (Eq. (2) in the main text):

$$q(\boldsymbol{\xi}_b) = \mathrm{Categorical}\left(\boldsymbol{\xi}_b \middle| \Pi_b(\boldsymbol{x})\right), \ \forall b, \quad \boldsymbol{\Pi}_b(\boldsymbol{x}) = \mathrm{Softmax}\left(\sum_{j=1}^{J}[w_{j,b,u}]_{u=1}^{U} \cdot x_j\right) \quad (5)$$

where $\Pi_b(\boldsymbol{x})$ is a vector comprising the *activation probability* of each neuron in the block, and $[w_{j,b,u}]_{u=1}^{U}$ denotes the vector concatenation of the set $\{w_{j,b,u}\}_{u=1}^{U}$.

---

[1] https://github.com/pytorch/examples/tree/main/imagenet
[2] https://github.com/facebookresearch/deit.

Then, the samples $\hat{\boldsymbol{\xi}}$ are expressed as:

$$\hat{\xi}_{b,u} = \mathrm{Softmax}\left(\log[\mathbf{\Pi}_b(\boldsymbol{x})]_u + g_{b,u})/\tau\right), \ \forall b = 1, \dots, B, \ u = 1, \dots, U \tag{6}$$

where $g_{b,u} = -\log(-\log V_{b,u})$, $V_{b,u} \sim \mathrm{Uniform}(0,1)$, and $\tau \in (0, \infty)$ is a temperature factor, controlling how "closely" the continuous relaxation approximates the Categorical distribution. In this work, we use a temperature of $\tau = 0.67$ as suggested in [16], and used in several other works [22, 21].

### A.3 Convolutional Formulation

In this setting, local competition is performed among feature maps on a *position-wise* basis. Each kernel is treated as an LWTA block with competing feature maps; each layer comprises $B$ kernels. Specifically, each feature map $u = 1, \dots, U$ in the $b^{\mathrm{th}}$ LWTA block of a convolutional LWTA layer computes:

$$\boldsymbol{H}_{b,u} = \boldsymbol{W}_{b,u} \star \boldsymbol{X} \in \mathbb{R}^{H \times L} \tag{7}$$

Competition remains stochastic, and is now implemented on a *position-wise* basis as follows:

$$q(\boldsymbol{\xi}_{b,h',l'}) = \mathrm{Categorical}\left(\boldsymbol{\xi}_{b,h',l'} \,\middle|\, \mathbf{\Pi}_{b,h',l'}(\boldsymbol{X})\right), \ \forall h', l' \tag{8}$$

where $\mathbf{\Pi}_{b,h',l'}(\boldsymbol{X}) = \mathrm{Softmax}\left([\boldsymbol{H}_{b,1,h',l'}, \dots, \boldsymbol{H}_{b,U,h',l'}]\right)$ comprises the *position-wise activation probabilities* for all neurons in the block.

For each position in a kernel, only the feature map that wins the said position contains a non-zero entry. This yields sparse feature maps with mutually exclusive activated positions. Now, the output $\boldsymbol{Y} \in \mathbb{R}^{H \times L \times B \cdot U}$ is obtained via concatenation of the sub-tensors $\boldsymbol{Y}_{b,u}$ that read:

$$\boldsymbol{Y}_{b,u} = \boldsymbol{\Xi}_{b,u}\left(\boldsymbol{W}_{b,u} \star \boldsymbol{X}\right), \ \forall b, u \tag{9}$$

where $\boldsymbol{\Xi}_{b,u} = [\xi_{b,u,h',l'}]_{h',l'=1}^{H,L}$. The corresponding detailed bisection of a convolutional stochastic LWTA block can be found in the Appendix.

In the following, we use these definitions to construct Competitive Vision Networks by replacing the usually employed non-linearities with the competition mechanism in each hidden layer of the considered architecture.

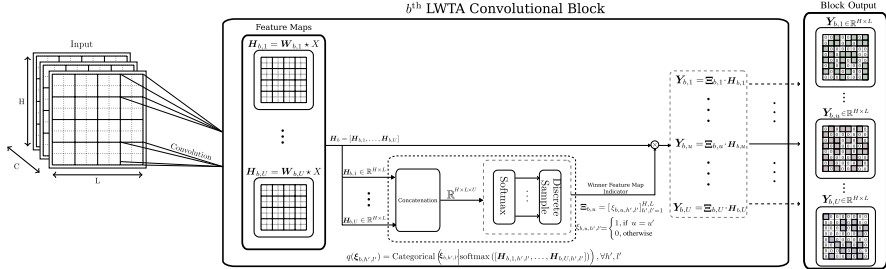

Figure 4: A detailed bisection of the $b^{\mathrm{th}}$ convolutional stochastic LWTA block. Presented with input $\boldsymbol{X} \in \mathbb{R}^{H \times L \times C}$, competition takes place among feature maps on a position-specific basis. Only the winner feature map contains a non-zero entry in a specific position. This leads to sparse feature maps, each comprising uniquely position-wise activated pixels. Image from [23].

## B  Further Qualitative Analysis

In this section, we provide further qualitative neuron identification results for various architectures and similarity functions.

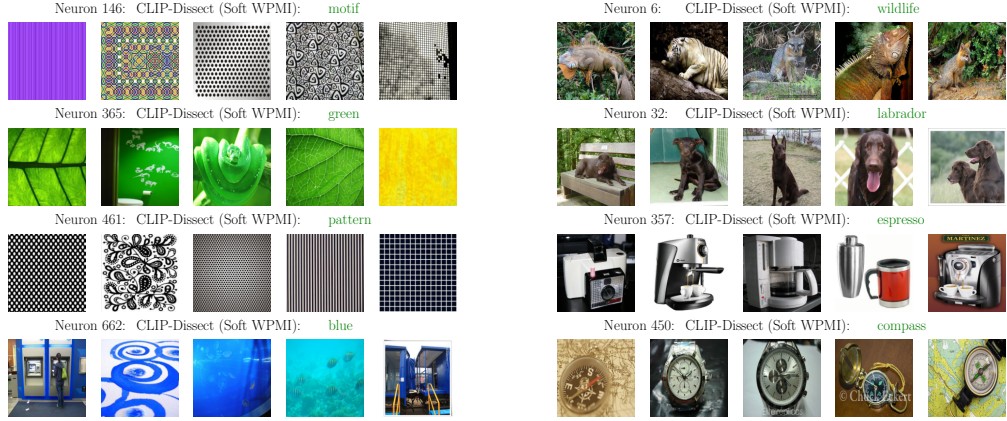

Figure 5: Neuron Identification for the first and last DeiT-T/8 MLP blocks using: SoftWPMI, $\mathcal{D}_{\text{probe}}$: ImageNet & Broden, $\mathcal{S}$: $20K$ most common English words[20].

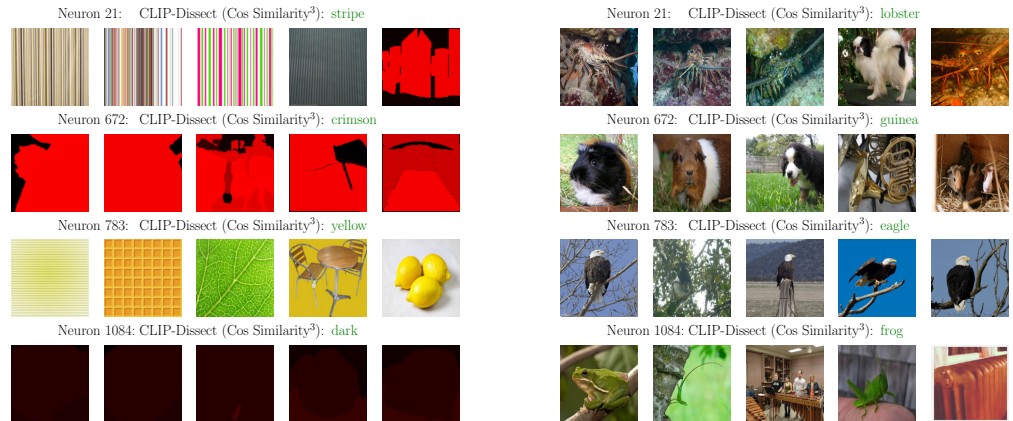

Figure 6: Neuron Identification for the first and last DeiT-S/12 MLP block using the Cosine Similarity Cubed similarity function [20]. For the former we use $\mathcal{D}_{\text{probe}}$: (i) ImageNet Val & Broden (Left), and for the latter: (ii) ImageNet Val. We use the $\mathcal{S}$: $20K$ most common English words for both settings.

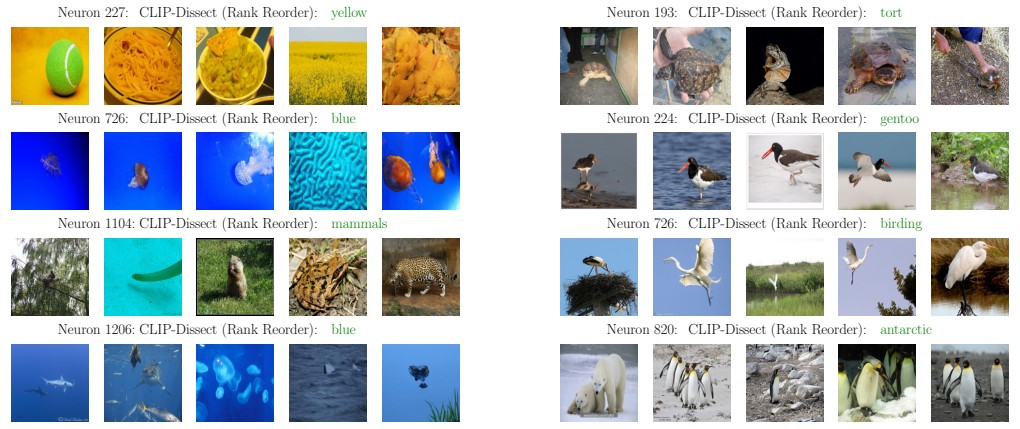

Figure 7: Neuron Identification for the first and last DeiT-S/12 MLP block using the Rank Reorder similarity function [20]. We use $\mathcal{D}_{\text{probe}}$: ImageNet Val and $\mathcal{S}$: $20K$ most common English words for both settings.

Table 5: Concepts tied to activated neurons for a random image from the ImageNet validation set for DeiT-T/8 using various similarity functions.

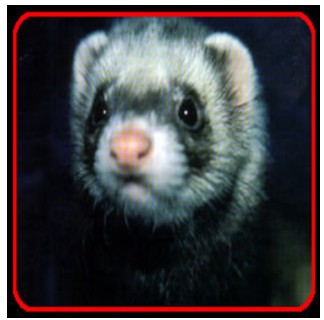

| Cos Similarity | | WPMI | | SoftWPMI | |
| --- | --- | --- | --- | --- | --- |
| Concepts | Act | Concepts | Act | Act | Concepts |
| Belgian Malinois | +10.5 | ferret | +10.5 | ferret | +10.5 |
| green/yellow eyes | +9.55 | ferret | +9.55 | ferret | +9.55 |
| white fur underside | +8.90 | white fur under tail | +8.90 | white fur under tail | +8.90 |
| herd Alpine ibex | +8.51 | black stripes on legs | +8.51 | black/white stripes | +8.51 |
| long/sharp quills | +8.02 | long/sharp quills | +8.01 | long/sharp quills | +8.01 |
| gray fur/white tips | +7.53 | fox | +7.53 | fox | +7.53 |
| white/gray head | +7.52 | white/gray head | +7.52 | white/gray head | +7.52 |
| baby stork | +0.96 | black ruff around neck | +1.09 | black ruff around neck | +1.09 |
| black fur/white markings | +0.44 | black/tan coloration | +0.44 | black/tan coloration | +0.44 |
| large/fluffy white dog | +0.35 | organ | +0.37 | organ | +0.37 |
| gills | +0.18 | salamander | +0.18 | salamander | +0.18 |
| car towing RV | +0.12 | headsail | +0.12 | headsail | +0.12 |
| other crocodiles | +0.12 | egg-laying mammal | +0.12 | egg-laying mammal | +0.12 |

Table 6: Concepts tied to activated neurons for a random image from the ImageNet validation set for DeiT-T/None and DeiT-S/12 using various similarity functions.

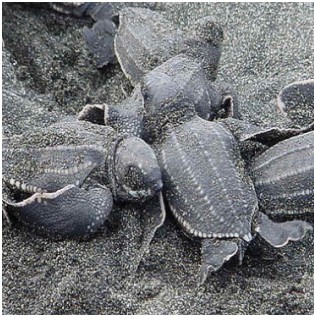

| DeiT-T/None (SoftWPMI) | | DeiT-S/12 (SoftWPMI) | | DeiT-S/12 (Rank Reorder) | |
| --- | --- | --- | --- | --- | --- |
| Concepts | Act | Concepts | Act | Act | Concepts |
| a coast | +1.18 | sea turtle | +15.0 | sea turtle | +15.2 |
| hard/shiny exoskeleton | +1.06 | cygnet | +11.7 | large/webbed hind feet | +11.7 |
| sea turtle | +0.54 | large/elephant animal | +10.9 | dark-colored carapace | +10.9 |
| large/red crab | +0.54 | hard shell covered in spines | +9.96 | large/fleshy cap | +9.96 |
| large/red crustacean | +0.53 | sea turtle | +9.52 | marine ecosystem | +9.52 |
| large/plant eating dinosaur | +0.51 | python | +9.32 | tough/scaly exterior | +9.32 |
| short/bristly fur | +0.46 | doberman | +9.08 | black/tan coloration | +9.08 |
| white bill | +0.20 | dome-shaped cap | +1.14 | round/spouted body | +1.41 |
| a race | +0.17 | iguana | +0.63 | fluffy/white appearance | +0.33 |
| large/fluffy white dog | +0.06 | other crocodiles | +0.67 | operating system | +0.22 |
| two-metal plates/trays | −0.16 | case | +0.22 | opponent | +0.10 |
| white markings wings | −0.16 | players | +0.10 | wings attached | +0.10 |
| small/crab like body | −0.17 | exoskeleton | +0.01 | hard/shiny exoskeleton | +0.10 |

