# OpenReview forum: "DISCOVER: Making Vision Networks Interpretable via Competition and Dissection"
_NeurIPS.cc/2023/Conference — NeurIPS 2023 poster_

### Official Review · Reviewer_PAgG · 2023-07-06

**Soundness:** 3 good
**Presentation:** 3 good
**Contribution:** 2 fair
**Rating:** 6
**Confidence:** 2

**Summary:**

This paper proposes a Dissection of Competitive Vision Networks (DISCOVER) to build interpretable neural networks. It aims to understand the neuron functionality for image inference. Moreover, this paper introduces the Jensen-Shannon divergence between the probability of concept presence and the probability of neuron activation for network dissection. Experimental results evaluate the effectiveness of the proposed CVNs.

**Strengths:**

1. Relatively interesting idea of the proposed CVNs for network dissection.
2. high interpretability of the proposed framework evaluated in experiments.


**Weaknesses:**

1. Limited novelty. DISCOVER method is similar to DMoE [ref-1] combined with the work [ref-2]. Please clarify their difference.
2. Lack of more visualization results to evaluate the interpretability over other methods.

[Ref-1]. Deep mixture of experts via shallow embedding. Uncertainty in artificial intelligence. PMLR, 2020.
[Ref-2] Interpretable Neural Network Decoupling. In ECCV, 2020.


**Questions:**

Refer to the Weaknesses.

**Limitations:**

Yes.

---

> ### Author Rebuttal · Authors · 2023-08-08
>
> We thank the reviewer for kind comments concerning our approach and for pointing out potentially related work to ours. We believe however that the reviewer is a bit harsh concerning the novelty of the proposed framework. We support this claim by highlighting the pivotal differences between our work and the suggested works.
>
> > Limited novelty. DISCOVER method is similar to DMoE [ref-1] combined with the work [ref-2]. Please
> clarify their difference.
>
> Both DMoE [ref-1] and the proposed Competitive Vision networks aim to sparsify the networks on a per example basis and do so in a way that renders the networks more expressive. However, the setup of the networks, the rationale, the training procedures and overall the way these two  methods achieve this sparsification is radically different. On the one hand, in DMoE, softmax-ed embeddings of the *input image* are obtained using a "shallow" network; these are then passed through a RELU activated multi-headed sparse gating networks to sparsify the results in each layer. On the other hand, CVNs group feature maps into blocks and a competitive random sampling procedure is employed in order to select the winner in each block. This is instantiated by using the response of each feature map to the **layer input** (and not the original image) and by introducing appropriate discrete latent indicators to select the winners that are modeled via solid Bayesian arguments.
>
> Moreover: (i) the "shallow" embedding network that DMoE introduces comprises 4-5 layers; in contrast we do not introduce any additional parameters and use the layer embedding of its input to drive the sparsification mechanism. At the same time the embeddings of DMoE not only arise from the original raw image, but a different weight matrix is introduced for each layer via the gating network that multiply the latent mixture weights, while relying on the ReLU operation to "sparsify" the results; instead, we rely on foundational Bayesian arguments and procedures to learn categorical variables that explicitly indicate unit activation in a competitive manner.
>
> With respect to [ref-2], their method uses a Mutual Information based approach between the inputs and an encoding vector in each layer, in order to discover a unique calculation path for each input. Similarly to the previously discussed work, the only similarity this method bears with our CVN models is the per-example sparsification of the network. We do not consider any mutual information constraints, we do not use any Architecture Controlling Modules, i.e., additional ad-hoc embedding networks to drive the sparsification and more importantly we rely on the concept of local competition between channels, instantiated via a principled formulation via appropriate discrete latent variables and not discretization approaches for training binary (and not categorical) latent variables.
>
>
> > Lack of more visualization results to evaluate the interpretability over other methods.
>
> As mentioned in the main text, line 305, and our responses to the concerns of other reviewers, since we trained the novel CVN models from scratch, and considering the highly different mode of operation, a direct comparison with other inteprertable methods is not feasible at this point. In the context of this work, and since interpretability in this case is based on the novel CLIP-Dissect framework, most visualization pertain to the standalone Neuron Identification results and the relative to conventional networks quantitative analysis provided in the main text (with some additional qualitative figures in the Supplementary); we nevertheless showed that CVNs improve performance compared to the their conventional counterparts. Since this is the first work that trained CVNs on a large scale, we aim to expand the interpetability investigation to other interpretability-focused methods.

---

> > ### Comment · Reviewer_PAgG · 2023-08-18
> >
> > Thanks for your rebuttal. The responses are well addressed my concerns. I have no more questions. However, I am not familiar to this area. So I have a low confidence to this paper.

---

> > > ### Author Response · Authors · 2023-08-20
> > >
> > > We thank the reviewer for acknowledging our efforts during the rebuttal period. We will add the discussion for the suggested related work in the final version to highlight the differences.

---

### Official Review · Reviewer_CPpE · 2023-07-07

**Soundness:** 3 good
**Presentation:** 4 excellent
**Contribution:** 3 good
**Rating:** 7
**Confidence:** 3

**Summary:**

This paper proposes to use stochastic local winner-takes-all (LWTA) layers in vision networks combined with CLIP-Dissect to improve the ability of a neuron to capture a concept. A stochastic LWTA layer groups pre-activation neurons into $B$ blocks of $U$ features each and within each block, chooses one feature with probability according to the softmax of the pre-activation neurons in the block. The chosen feature is preserved while the rest are zeroed out.

The paper tests DeiT-T, DeiT-S, and ResNet-18 architectures on ImageNet and Places365 replacing original activation functions with stochastic LWTA. The classification performance slightly improves when each block is of size 2, or $U=2$, and degrades as $U$ increases.

The stochastic nature of LWTA allows the authors to use CLIP-Dissect with the Jensen-Shannon Divergence (JSD), which they do. The learned concepts for the different investigated neurons, such as “green,” “red,” “trains,” and “snowmobiles” match the images those neurons are activated for. The text embedding similarity of found concepts to ground truth ImageNet label is higher with LWTA neurons and JSD than normal neurons and WPMI.

**Strengths:**

The presentation is clear and the motivation in the opening sections is well-expressed. Also, extensive code is provided to support the approach.

The idea of introducing competition between neurons and having one win out seems intuitive and suited for interpretability, the stated goal of this paper.

The results are promising (the introduced module does not harm performance too much, and potentially increases performance if a very mild form of it with $U=2$ is used, and some quantitative evidence for superior interpretability is provided).

**Weaknesses:**

It’s not fully clear whether the approach is being presented as an ante-hoc method or a post-hoc method. It seems one could argue that needing to train a network from scratch with the LWTA nonlinearity makes it an ante-hoc method. It also seems one could argue that since LWTA is a drop-in replacement for any non-linearity, it functions more as an enhancement to any ante-hoc or post-hoc methods applied on top of the architecture rather than an ante-hoc/post-hoc method itself. Could the authors clarify this?

The evaluation could be more thorough and more convincing. Here are some suggestions, not all of which might be necessary:
- Significant decline in performance is stated as a drawback of other ante-hoc methods, which this method is arguably an instance of; is it possible to compare classification performance with one or more baseline ante-hoc approaches?
- Can you provide some sort of comparison between $U$ and the interpretability of the proposed approach? Such as, a plot of $U$ vs. cosine similarity between intermediate neuron description, or comparing concepts found with $U=2$ to $U=16$. Since $U=2$ achieves the best performance, it’s not clear why we would prefer $U=16$. The argument that $U=16$ achieves higher sparsity and thus higher interpretability intuitively makes sense, but is this supported empirically?
- In my opinion, an additional interpretability metric to evaluate the potential of CVNs for interpretability purposes would improve the paper. Would it make sense to use the IOU-based interpretability metric from the Network Dissection paper? If that would require too much additional experimentation, is some sort of more lightweight added experiment/comparison possible?

Miscellaneous typos/comments:
- L177: Doesn’t $\mathrm{CE}(y_i, f(X_i, \hat\xi))$ denote the cross entropy, not $f(X_i, \hat\xi)$
- L340: “the resulting networks yielded on par or even improved performance, even when using up to 4% of active neurons per example” seems a bit of a stretch; going from original network to most sparse competition-based network for DeiT-T on ImageNet-1k, the accuracy dropped from 72.2 to 70.5, for ResNet18 on Places365, from 53.5 to 49.5. I would call these drops significant. I would call the drop of DeiT-S on ImageNet-1k from 77.0 to 76.7 “on par”, but not improved.

**Questions:**

I put the most relevant questions in the “Weaknesses” section.
- In L228, it is claimed “In the context of inner-product neurons, we directly obtain the presence probability via Eq. (2).” But aren’t vision transformer neurons also spatially arranged, like convolutional neurons? I.e. for a given feature there are different instances of it which corresponding to a spatial map, corresponding to the different embedded patches. So do you also average over spatial positions for transformer?

**Limitations:**

Limitations were adequately addressed.

---

> ### Author Rebuttal · Authors · 2023-08-08
>
> We thank the reviewer for his insightful analysis and positive comments for our work. We will correct and update the phrasing on the noted typos/comments for the camera-ready.
>
> > It’s not fully clear whether the approach is being presented as an ante-hoc method or a post-hoc method.
>
> This constitutes a very interesting point and we'll clarify in the camera ready. Post-hoc interpretability methods are considered for already trained backbone architectures, where a post-hoc analysis method like Neuron Dissection is applied. In the case of LWTA-based networks, there weren't any pretrained models available to apply Network Dissection on. That is why we need it to train them from scratch; after publication, these will be made available for further research investigations. Thus, the proposed framework constitutes an application of a post-hoc interpretability method to CVNs since it is based on the CLIP-Dissect approach that is itself post-hoc; nevertheless it required training since pretrained models were not available.
>
> The LWTA mechanism is easily applicable to any model architecture, however it can't just be used as drop-in replacement to existing ReLU-trained backbones. For example, replacing the activations in a pretrained ReLU network and directly performing inference will lead to performance degradation. This is natural considering the LWTA unique mode of operation. The model needs to be trained with the competition mechanism to learn the different neuronal activation patterns and how the grouped components compete for their output. In this work, we argue and experimentally validate that this mechanism is indeed a viable alternative to ReLU-based activations, that is accompanied by some significant benefits in term of per-example activation sparsity, and subsequently, intepretability.
>
> > Significant decline in performance is stated as a drawback of other ante-hoc methods, which this method is arguably an instance of; is it possible to compare classification performance with one or more baseline ante-hoc approaches?
>
> We believe that the answer to the previous question clarifies that our approach is not an ante-hoc method like Concept Bottleneck Models. We are simply training Vision Networks from scratch with the LWTA competition mechanism instead of the conventional non-linearities. Moreover, the obtained performance is comparable or even improves the classification results compared to the baseline conventional networks in some settings.
>
> > Can you provide some sort of comparison between $U$ and the interpretability of the proposed approach? The argument that achieves higher sparsity and thus higher interpretability intuitively makes sense, but is this supported empirically?
>
>  It is true that $U=2$ offers the best-performance accuracy-wise under the same training regime but in the scope of per-example sparsity $U=16$ offers a better alternative. However, this question is related to Question 3 of Rev. BNcZ, where we noted that:
>
>     In the context of interpretability, activating, e.g., 1/16 of the original neurons per each layer, surely facilitates the process of neuron specialization: since not all neurons are activated for each example, neurons can specialize to only the features of the examples for which they are active. After Neuron Identification is performed, we can even narrow down the neurons that were activated by a particular test input and try to reason using only these for a downstream task. For example, if we consider a hidden layer with 800 components and U=16 competitors, it is easier to reason and investigate the 50 activated components instead of investigating the effect all components in conventional architectures.
>
> The per-example sparsity will facilitate the examination and potentially correction of the networks decisions. In the context of the used similarity metrics, we obtain similar cosine similarity values for different $U$ values. Our next steps comprise the exploration of the activation patterns and interpretation results in various settings. This nevertheless requires extensive investigations that go beyond the scope of this work.
>
> >  In my opinion, an additional interpretability metric to evaluate the potential of CVNs for interpretability purposes would improve the paper. Would it make sense to use the IOU-based interpretability metric from the Network Dissection paper? If that would require too much additional experimentation, is some sort of more lightweight added experiment/comparison possible?
>
> This is an interesting suggestion from the reviewer. However, considering other interpretability metrics as the IOU one proposed in the Network Dissection paper 1) requires a totally different treatment, and (2) defeats the purpose of using the post-hoc CLIP-DISSECT method. In this work, we selected the CLIP-DISSECT method over the conventional NetworkDissection. That is why even in the original CLIP-DISSECT publication there isn't any comparison on a metric basis using the IOU or other interpretability measures proposed in related works.
>
> > In L228, it is claimed "In the context of inner-product neurons, we directly obtain the presence probability via Eq. (2)." But aren’t vision transformer neurons also spatially arranged, like convolutional neurons? I.e. for a given feature there are different instances of it which corresponding to a spatial map, corresponding to the different embedded patches. So do you also average over spatial positions for transformer?
>
> We thank the reviewer for raising this point. It is something that we wanted to clarify in the main text. For the transformer architecture in particular, we alter the MLP block of the transformer architecture and specifically we replace the used GeLU architecture with the LWTA mechanism. Indeed in this case, we average over the spatial positions in order to obtain the probability.

---

> > ### Comment · Reviewer_CPpE · 2023-08-20
> >
> > I have read the authors' response and stand by my initial evaluation.
> >
> > It remains a point of possible improvement that the authors provide a concrete example of how a sparser CVN (CVN trained with higher $U$) provides better interpretability. The intuitive argument makes sense, but a case study would really help drive it home. I am curious what would happen if for a couple images, you listed all of the concepts corresponding to the activated neurons -- this is something the CVN design is particularly suited for as the latent one-hots explicitly declare which neurons are activated. Would CVNs with $U=2$ have a lot more irrelevant concepts than CVNs with $U=16$? That would be interesting to show.

---

> > > ### Author Response · Authors · 2023-08-20
> > >
> > > We are glad that through this discussion, we were able to reach an agreement that in this setting the sparse activation argument makes sense. Moreover, the argument that the reviewer is making, i.e., that using the latent one-hot vectors to investigate which neurons are activated for each example (or even find patterns of activations among different images), is exactly one of the most important properties of CVNs. This will be a very important addition to our work. We are currently working on this analysis; we'll do our utmost to provide the results before the end of the discussion period.

---

> > > > ### Author Response · Authors · 2023-08-21
> > > >
> > > > We thank the reviewer for his patience.
> > > > We have now obtained an analysis of the neuron activations for a random example from the ImageNet-1k validation set. Since we cannot post links to external documents, we will include all the information in this and the following comment.
> > > >
> > > > The randomly chosen example is the 9534th example of the validation set, depicting a showdog with white fur and some other light colors. The dog is upon a metallic table with wheels beside some prizes and with a human behind shown until the chest. There is a small fence in the picture, of the color yellow/orange, a wood wall in the background and some other elements. This example can be loaded and visualized through our submitted code by:
> > > >
> > > > ```
> > > > import data_utils
> > > > import matplotlib.pyplot as plt
> > > >
> > > > dprobe = 'imagenet_val'
> > > > pil_data = data_utils.get_data(dprobe)
> > > > im, label = pil_data[9534]
> > > > im = im.resize([375,375])
> > > > plt.imshow(im)
> > > > plt.axis('off')
> > > > plt.show()
> > > > ```
> > > >
> > > > For this image, we compute the responses of the neurons for the last MLP layer after the application of the activation function (GeLU or LWTA). We consider the conventional GeLU-based DeiT network and a CVN with either 8 or 16 competitors.
> > > >
> > > > For the conventional architecture we found that 766 out of the 768 neurons are active with concepts such as:
> > > >
> > > > >a condenser,
> > > > synthetic or real hair,
> > > > a bowl-shaped top,
> > > > a green body with brown spots,
> > > > a soft, wheaten-colored coat,
> > > > camels,
> > > > a large front grill,
> > > > a moving truck,
> > > > a ski helmet,
> > > > a small, black dog,
> > > > a whiptail lizard,
> > > > a ribbon or string tie,
> > > > a golf cart driver,
> > > > a whiptail lizard,
> > > > a third row of seating,
> > > > a small, round fruit,
> > > > a tall, cylindrical chimney,
> > > > a urn,
> > > > string instrument,
> > > > echinoderm,
> > > > a strap or band around the hat,
> > > > a father duck,
> > > > on a plant,
> > > > a herding dog,
> > > > a juicy interior with seeds,
> > > > a large selection of books,
> > > > supported by columns or arches,
> > > > red or wheaten color,
> > > > a safety vest,
> > > > holding a baseball bat,
> > > > sea turtle,
> > > > a herding dog,
> > > > a father bird,
> > > > a mottled brown plumage,
> > > > bicycles,
> > > > a travel bag,
> > > > two metal plates or trays,
> > > > an easel,
> > > > a coral,
> > > > a beetle,
> > > > a train,
> > > > a blue-gray body,
> > > > a bipod,
> > > > a white and liver-colored coat,
> > > > bathroom,
> > > > organ,
> > > > a fax machine,
> > > > a cassette slot,
> > > > worn by soldiers and knights,
> > > > a herd of wild boar,
> > > > bacteria,
> > > > large, triangular fins,
> > > > white fur under the tail,
> > > > a counter or bar area,
> > > > a fare box,
> > > > a gas-filled interior,
> > > > a stitched design,
> > > > a white and liver-colored coat,
> > > > a TV stand,
> > > > lagomorph,
> > > > garden equipment,
> > > > measurement device,
> > > > a small, shaggy dog,
> > > > a penguin colony,
> > > > brace,
> > > > in water,
> > > > a ski helmet,
> > > > a reddish-brown body,
> > > > a seed drill,
> > > > a paintbrush holder,
> > > > stitched or appliqued design,
> > > > a sleeveless garment,
> > > > a large, red crustacean,
> > > > a cable box,
> > > > a label with the product name,
> > > > a mottled brown plumage,
> > > > white markings on the wings,
> > > > a military vehicle,
> > > > a ski helmet,
> > > > a white bill,
> > > > a plane,
> > > > a car towing a RV,
> > > > a comic book reader,
> > > > a black ruff around the neck,
> > > > farm equipment,
> > > > a mottled brown plumage,
> > > > a garment bag,
> > > > a speaker stand,
> > > > a camera,
> > > > a fox,
> > > > red antennae,
> > > > shelves inside,
> > > > NASA markings,
> > > > a drilling site,
> > > > a donkey,
> > > > a club-shaped pestle,
> > > > a long, flat expanse of sand,
> > > > a chicken,
> > > > a set of tuned wooden bars,
> > > > a beverage fridge,
> > > > a stovetop,
> > > > a scanner,
> > > > a gun belt,
> > > > a pool table brush,
> > > > a person eating a pretzel,
> > > > other siamangs,
> > > > numeric keypad,
> > > > on a plant,
> > > > a man wearing a tuxedo,
> > > > a spotted coat,
> > > > a fox,
> > > > a tall, slender cabinet,
> > > > a web-like structure,
> > > > a white and liver-colored coat,
> > > > ping-pong paddles,
> > > > a bus station,
> > > > a decanter,
> > > > a bowl-shaped top,
> > > > other lemurs,
> > > > a small, forked tail,
> > > > a central skylight,
> > > > a hose or nozzle attachment,
> > > > a small, spiral shell,
> > > > large, transparent wings,
> > > > a white rump,
> > > > a downspout,
> > > > brake fluid reservoir,
> > > > a bed frame,
> > > > a metal plaque, ...
> > > >
> > > > These are 131 out of the 766 activated concepts and we omit the rest for brevity.

---

> > > > > ### Author Response · Authors · 2023-08-21
> > > > >
> > > > > In the case of a CVN with $8$ competitors, we yield $96$ activated neurons out of the $768$ as expected $ (768/8)$.
> > > > >
> > > > > > a butterfly bush,
> > > > > a peacock,
> > > > > a soft, wheaten-colored coat,
> > > > > a measuring tape,
> > > > > a sleeveless garment,
> > > > > other langurs,
> > > > > a penguin colony,
> > > > > a self-contained living space,
> > > > > a shaggy mane,
> > > > > a croquet set,
> > > > > a zeppelin,
> > > > > a small case or box,
> > > > > Spaniel,
> > > > > kernels inside the shell,
> > > > > a rabbit,
> > > > > a club-shaped pestle,
> > > > > a mother koala,
> > > > > a crosswalk,
> > > > > long, sharp quills,
> > > > > a wolf-like appearance,
> > > > > a small, shaggy dog,
> > > > > black and white stripes,
> > > > > used for seeing behind the car,
> > > > > a herd of Alpine ibex,
> > > > > a QWERTY layout,
> > > > > a tropical fruit,
> > > > > a small, mushroom-like shape,
> > > > > a soft, wheaten-colored coat,
> > > > > citrus fruit,
> > > > > a TV remote,
> > > > > a shaggy, black and brown coat,
> > > > > a log pond,
> > > > > a ski helmet,
> > > > > short, black-and-white legs,
> > > > > a shaggy, gray coat,
> > > > > a large, coiled shell,
> > > > > a Savannah,
> > > > > a rufous back and wings,
> > > > > a black cap and bib,
> > > > > a black, brindle, or fawn coat,
> > > > > a long, pointed bill,
> > > > > a large dish or antenna,
> > > > > a tropical fruit,
> > > > > a Savannah,
> > > > > a parachute pack,
> > > > > a marsupial pouch,
> > > > > puzzle,
> > > > > a large, elephant-like animal,
> > > > > jug,
> > > > > a rufous back and wings,
> > > > > a large, colorful bird,
> > > > > white markings on the wings,
> > > > > hippos,
> > > > > a large, colorful bird,
> > > > > large, transparent wings,
> > > > > a bike,
> > > > > a shaggy, black and brown coat,
> > > > > a QWERTY layout,
> > > > > a variety of pendants or beads,
> > > > > a herd of Alpine ibex,
> > > > > a colorful, wheel-shaped toy,
> > > > > egret,
> > > > > a parachute pack,
> > > > > a large dish or antenna,
> > > > > hippos,
> > > > > a large, colorful bird,
> > > > > thick, cabbage-like leaves,
> > > > > a van or truck chassis,
> > > > > a long head with erect ears,
> > > > > a checkered or solid red sauce,
> > > > > a spinning stool,
> > > > > a mother koala,
> > > > > a male monkey,
> > > > > a large, adjustable headrest,
> > > > > a court,
> > > > > a soft, wheaten-colored coat,
> > > > > a small, oval-shaped bed,
> > > > > a tropical fruit,
> > > > > a large, plant-eating dinosaur,
> > > > > a short, reddish brown coat,
> > > > > a lock,
> > > > > Spaniel,
> > > > > ping-pong paddles,
> > > > > a white or grey head,
> > > > > a white and liver-colored coat,
> > > > > large, triangular fins,
> > > > > lemur,
> > > > > a soft, wheaten-colored coat,
> > > > > a large, fluffy white dog,
> > > > > a paddleboard,
> > > > > a tall, pink bird,
> > > > > a portafilter,
> > > > > a large, fluffy white dog,
> > > > > a travel bag,
> > > > > a bridge connecting the lenses,
> > > > > a pumpkin shape.
> > > > >
> > > > > This is the whole list of concepts activated for this particular example. It is expected that some of the activated neurons will be irrelevant to for describing the input example. However, we have managed to narrow down the list of concepts by a factor of 8, while retaining highly relevant neuron activation such as *a large, fluffy white dog, a white or grey head, a small, shaggy dog, e.t.c.*
> > > > >
> > > > > Finally, when considering $U=16$ competitors, we yield **a total of** $48$ activated neurons with concepts:
> > > > >
> > > > > > a thick, shaggy coat,
> > > > > a three-hulled vessel,
> > > > > a shaggy mane,
> > > > > kernels inside the shell,
> > > > > stitched or appliqued design,
> > > > > a high chair,
> > > > > fore-and-aft sails,
> > > > > a spinning stool,
> > > > > other langurs,
> > > > > a fossil,
> > > > > a small, white dog,
> > > > > a shaggy, gray coat,
> > > > > a soft, wheaten-colored coat,
> > > > > a black-and-tan coat color,
> > > > > heron,
> > > > > a white or cream-colored coat,
> > > > > a large, fluffy white dog,
> > > > > a small, brown shell,
> > > > > fastener,
> > > > > a reddish-brown body,
> > > > > a small, shaggy dog,
> > > > > instrumentalists,
> > > > > a white and liver-colored coat,
> > > > > chopped vegetables,
> > > > > a terrier,
> > > > > a large, upholstered chair,
> > > > > medical instrument,
> > > > > a person in a lab coat,
> > > > > a shaggy, gray coat,
> > > > > a white and liver-colored coat,
> > > > > a slow-cooking function,
> > > > > a clog maker,
> > > > > red or wheaten color,
> > > > > a deep, bowl-shaped scoop,
> > > > > sea turtle,
> > > > > a series of beads on each rod,
> > > > > a large, elegant dog,
> > > > > a dark-colored carapace,
> > > > > a terrier,
> > > > > a terrier,
> > > > > a parade,
> > > > > a small, dainty dog,
> > > > > a small, spiral shell,
> > > > > a row of function keys,
> > > > > a whiptail lizard,
> > > > > a black-tipped tail,
> > > > > a small, spiral shell,
> > > > > organ.
> > > > >
> > > > > In this setting, the benefits of the proposed mechanism are much more apparent. We can now examine $48$ out of the potential $768$ concepts according to the neurons that were active. These results also hint at the specialization properties of the proposed mechanism. We observe that in this case, the neurons activated are tied to more relevant concepts, yielding descriptions as *terrier, small dainty dog, shaggy grey coat, e.t.c.'.
> > > > >
> > > > > We will include these results in the final version of the paper, along with other examples from the validation set.
> > > > > We thank the reviewer for suggesting and motivating this exploration.

---

> > > > > > ### Comment · Reviewer_CPpE · 2023-08-21
> > > > > >
> > > > > > The $U=16$ list of concepts seems like a reasonable improvement over the $U=8$ list and sheds some light on the potential benefit of sparse activations! I will raise my score to 7.

---

### Official Review · Reviewer_crQf · 2023-07-08

**Soundness:** 2 fair
**Presentation:** 2 fair
**Contribution:** 3 good
**Rating:** 4
**Confidence:** 3

**Summary:**

This paper proposes a novel method for interpreting vision networks by combining an architecture that incorporates stochastic local-winner-takes-all activations with the CLIP-DISSECT framework. The author makes use of the probabilistic interpretation of LWTA as a categorical distribution to derive a similarity measure for LWTA activations based on the Jenson Shannon Divergence. They show the benefit of their approach by comparing the discovered interpretations to ground truth labels for conventional and competitive (LWTA) networks.

**Strengths:**

The studied problem of interpretability is very important, the idea is well motivated, the background and related work is informative, and the writing is clear.
The accuracy of the LWTA adapted transformer is very encouraging and suggests the usefulness of competition independently of interpretability.
The improvements in automated neuron description, are encouraging.

**Weaknesses:**


Clarity could be improved: The figures are not very helpful. Especially fig 1 could be improved by simplifying it  (e.g. without showing individual neurons and weights). In my opinion Fig 2 is not necessary and could instead make space for a much needed figure to illustrate the DISCOVER framework.
The math feels a bit convoluted with too many new symbols introduced in each paragraph. Again, simplification could help understandability.
The paper spends quite time explaining how LWTA and DISCOVER can be used for convolutional networks. However, the main results of the evaluation focus on transformer architectures for which it is less clear how LWTA is incorporated (only replacing the ReLU inside the MLP block or also in the attention and residual branch?). I suggest switching focus and making conv architectures much less prominent.

The evaluation feels incomplete. The main evidence for the efficacy of DISCOVER are:
1. the qualitative results in Figure 3, which are only partially convincing. Eg. the "aquaculture" neuron seems to react to a highway, and the "dashboard" neuron mostly picks up on hifi equipment.
2. Table 2, which shows that, compared to non-LWTA-networks) the textual embedding of the discovered neuron descriptions is more similar to the corresponding ground truth label. However the CLIP cosine similarity score is hard to interpret and the reported gains seem mild.
For a paper that focuses on interpretability, I find the results hard to interpret. As it stands, the paper has me convinced that stochastic LWTA is interesting and should be studied further, but fails to convince me of the added value of DISCOVER over e.g. CLIP-dissect.

**Questions:**

How exactly is LWTA incorporated into the transformer?

**Limitations:**

The paper does discuss two relevant limitations.

---

> ### Author Rebuttal · Authors · 2023-08-08
>
> We thank the reviewer for his kind comments on the usefulness of our approach.
>
> > Clarity could be improved: The figures are not very helpful. Especially fig 1 could be improved by simplifying it (e.g. without showing individual neurons and weights). In my opinion Fig 2 is not necessary and could instead make space for a much needed figure to illustrate the DISCOVER framework. The math feels a bit convoluted with too many new symbols introduced in each paragraph. Again, simplification could help understandability. The paper spends quite time explaining how LWTA and DISCOVER can be used for convolutional networks. However, the main results of the evaluation focus on transformer architectures for which it is less clear how LWTA is incorporated (only replacing the ReLU inside the MLP block or also in the attention and residual branch?). I suggest switching focus and making conv architectures much less prominent.
>
> We thank the reviewer for his suggestions. Since LWTA-based networks are quite underexplored, we focused on the most used operations, i.e., inner-product and convolutions. Per the reviewer's suggestion we will simplify the figures and notation in the camera-ready and shift the focus to the Transformer architectures. As the reviewer correctly notes, in this work, we replaced the GeLU nonlinearity in the MLP layer of the transformer: this decision was motivated by recent works that have shown that transformer MLPs encode knowledge attributes [1,2,3]. We aim to explore other modifications in transformer architectures in the near future.
>
> [1] Damai Dai, Li Dong, Yaru Hao, Zhifang Sui, Baobao Chang, and Furu Wei. Knowledge neurons in pretrained transformers, ACL, 2022
>
> [2] Mor Geva, Roei Schuster, Jonathan Berant, and Omer Levy. Transformer feed-forward layers are key-value memories, EMNLP, 2021
>
> [3] Kevin Meng, Arnab Sen Sharma, Alex Andonian, Yonatan Belinkov, and David Bau. Mass editing memory in a transformer, ICLR, 2023
>
> > The qualitative results in Figure 3, which are only partially convincing. Eg. the "aquaculture" neuron seems to react to a highway, and the "dashboard" neuron mostly picks up on hifi equipment.
>
> Please see the response to Q4 of Rev. kXaB:
>
>     There are instances where both the conventional networks and the competitive vision networks do not provide the optimal results, e.g., in Fig. 3 Neuron 242; this result was purposely included in the Figure to highlight this fact. This is also case with the original CLIP-DISSECT work: therein, in Fig. 1 Neuron 683, some of the examples that highly activate the neuron have no relationship with the Neuron Identification label. In turn, this is why we considered the quantitative analysis in Table 2 using the metrics introduced in the CLIP-Dissect paper. Using the introduced metrics, it is  quite clear that using the LWTA mechanism yields a higher cosine similarity between the intermediate neuron descriptions and the ground truth labels compared to the conventional method that does not employ the LWTA mechanism; this illustrates the improvement in performance. Devising other quantitative methods for comparing conventional and LWTA-based Vision Networks, especially in the context of Network Dissection where direct comparison is rendered difficult, is a substantial research challenge that we aim to nevertheless address in the future.
>
>  In our view, it is not an easy task to construct a best-performing method in all settings and tasks. However, we believe that the LWTA mechanism and its unique mode of operation provide substantial benefits for constructing more interpretable networks.
>
> > Table 2, which shows that, compared to non-LWTA-networks) the textual embedding of the discovered neuron descriptions is more similar to the corresponding ground truth label. However the CLIP cosine similarity score is hard to interpret and the reported gains seem mild. For a paper that focuses on interpretability, I find the results hard to interpret. As it stands, the paper has me convinced that stochastic LWTA is interesting and should be studied further, but fails to convince me of the added value of DISCOVER over e.g. CLIP-dissect.
>
> We believe that the previous response answers this question. We want to emphasize that CLIP-Dissect is a post-hoc intepretability method that was applied on top of conventional vision architectures. In this work, we explored the potency of Competitive Vision networks by training such networks from scratch in a variety of settings and applying CLIP-Dissect as a post-hoc method. Thus, in our view, it is not a matter of if DISCOVER is better than CLIP-Dissect (since it uses CLIP-Dissect itself) but if the inherent competition mechanism facilitates improvement in post hoc analysis and interepretability of the results. Thus, and given that we are "limited'" by the CLIP-Dissect framework, we believe that our investigation has provided strong empirical evidence for the efficacy of CVNs compared to conventional architectures in this setting.
>
> >  How exactly is LWTA incorporated into the transformer?
>
> We believe the answer in Question 1 of the reviewer answers this question. We focused on altering the GeLU activation in the MLP block of the transformer with the LWTA mechanism.

---

> > ### Comment · Reviewer_crQf · 2023-08-18
> > **Response to rebuttal**
> >
> > I thank the authors for the clarifications. I understand now that advertising the usefulness of stochastic LWTA is part of the contributions of this paper. In particular, highlighting the fact that it can help with interpretability without sacrificing performance.
> > Because of that, I raise my score to borderline.

---

> > > ### Author Response · Authors · 2023-08-20
> > >
> > > We thank the reviewer for taking the time to comment on our clarifications and for raising his score. We will make sure that all the reviewer's suggestions are incorporated in the final version.

---

### Official Review · Reviewer_kXaB · 2023-07-13

**Soundness:** 2 fair
**Presentation:** 2 fair
**Contribution:** 1 poor
**Rating:** 2
**Confidence:** 5

**Summary:**

The paper proposes DISCOVER, a novel framework for creating interpretable vision networks. It utilizes stochastic Winner-Takes-All layers and multimodal vision-text models to uncover the specialization of each neuron. The framework achieves high activation sparsity, allowing for direct examination of concept representations. It also introduces a new similarity metric for network dissection based on Jensen Shannon Divergence. Experimental results demonstrate improved classification performance and provide a principled framework for text-based descriptions of neuron representations. The paper pioneers the training of LWTA networks on large datasets, yielding interpretable functionality with a small subset of neurons per datapoint.

**Strengths:**

The strengths of this paper include:

1. Novel Framework for Network Dissection: The paper introduces a novel framework, DISCOVER, for creating interpretable vision networks. It combines stochastic Winner-Takes-All layers and multimodal vision-text models, providing a unique approach to uncovering neuron specialization.

2. Improved Classification Performance: Despite not performing hyperparameter tuning, the DISCOVER framework achieves comparable classification performance.

3. Interpretable Similarity Metric: The paper introduces a new similarity metric based on Jensen Shannon Divergence to match neuron representations to concepts.

4. Pioneering LWTA Networks: The paper contributes to the literature by being one of the first to train large-scale LWTA networks using substantial datasets and architectures.

**Weaknesses:**

1. The WTA approach in this neural network framework draws inspiration from previous methods employed before the advent of deep learning. For instance, techniques like bag of words, Locality-constrained Linear Coding (LLC) (Wang et al., "Locality-constrained linear coding for image classification," CVPR, 2010) and a long etc, also involved neuron competition for activation.

2. It would be valuable to have a clearer understanding of the level of sparsity in the network without WTA. While the paper mentions that in a network without WTA sparseness is "input specific," providing some statistical analysis or comparisons of the sparseness level with non-WTA networks could provide insights into the necessity and effectiveness of the WTA mechanism.

3. The paper assumes that sparse representations lead to more interpretable units, but the connection between sparsity across neurons and human interpretable concepts needs further clarification. It would be beneficial to elaborate on the theoretical arguments or underlying assumptions supporting this claim. Additionally, providing empirical evidence or previous research that supports the notion that sparse representations indeed enhance interpretability would strengthen the argument.

4. The qualitative results provided in the paper are limited, particularly regarding the comparison between applying the WTA mechanism and not applying it. It would be insightful to include examples of units that are not interpretable when the WTA mechanism is not utilized. This comparison would help illustrate the added interpretability that the WTA approach brings to the network and provide a more comprehensive understanding of its impact on the model's performance.

5. The dependence on the interpretability similarity metric raises questions about the generalizability of the results. While the paper introduces a new similarity metric based on Jensen Shannon Divergence, it would be valuable to investigate the robustness of the findings using alternative metrics and visualization techniques. Demonstrating the consistency of the results across different evaluation methods would strengthen the validity and generalizability of the proposed framework for increased interpretability. Exploring the impact of alternative metrics and visualization techniques could provide a more comprehensive understanding of the interpretability achieved by the network.

By addressing these concerns, the paper can further enhance its contributions by providing more insights into the necessity and effectiveness of the WTA mechanism, establishing theoretical justifications, and expanding the qualitative analysis to better understand the interpretability aspect.

**Questions:**

As I mentioned earlier, there is a concern regarding the lack of clear evidence establishing a direct relationship between sparsity and interpretability. While the paper assumes that sparse representations lead to increased interpretability, it would be beneficial to provide results that demonstrate this relationship across different levels of sparsity, utilizing various interpretability measures. By systematically examining the impact of different sparsity levels on interpretability using diverse evaluation metrics, the paper could provide stronger evidence for the claim that sparsity enhances the interpretability of individual units. This analysis would contribute to a deeper understanding of the relationship between sparsity and interpretability and provide more robust support for the proposed framework.

**Limitations:**

I believe the limitation section could take into account the points I raised in weaknesses, as they were not mentioned.

---

> ### Author Rebuttal · Authors · 2023-08-08
>
> We thank the reviewer for his constructive comments on improving the presentation of the strengths of our work.
>
> > Similarity of WTA to Linear Coding (LLC).
>
> The LWTA approach does indeed bear some similarity with Locality-constrained linear coding relating to the grouping of features and selecting one when using hard VQ. However the construction, learning of the features and applicability of the approaches are highly different.
>
> >  It would be valuable to have a clearer understanding of the level of sparsity in the network without WTA. While the paper mentions that in a network without WTA sparseness is "input specific," providing some statistical analysis or comparisons of the sparseness level with non-WTA networks could provide insights into the necessity and effectiveness of the WTA mechanism.
>
> This is a very valuable suggestion from the reviewer and will be included in the main text.
>
> We consider the official pretrained conventional DeiT-T checkpoint with the standard GeLU activation and report the sparsity level: this is obtained by averaging the number of activated neurons after the application of the GeLU nonlinearity in the MLP block of DeiT and considering each neuron as active if its absolute value is greater than a small constant, i.e., $10^{-6}$ to account for rounding errors. For this test, we average over all the test examples of the ImageNet-1k validation set in all layers of DeiT. We observed that $98$% of the neurons are active in this case. Using the exact same implementation for our LWTA-based DeiT-tiny implementation with 16 competitors, yields the expected $6.25$% sparsity.
>
> > The paper assumes that sparse representations lead to more interpretable units, but the connection between sparsity across neurons and human interpretable concepts needs further clarification.
>
> We thank the reviewer for raising this point as it will help improve on the phrasing of the main text and avoid potential misunderstandings for our claims about the relation between sparsity and interpretability. For this point, we re-iterate the response to Question 3 of reviewer BNcZ:
>
>     In the context of interpretability, activating, e.g., 1/16 of the original neurons per each layer, surely facilitates the process of neuron specialization: since not all neurons are activated for each example, neurons can specialize to only the features of the examples for which they are active. After Neuron Identification is performed, we can even narrow down the neurons that were activated by a particular test input and try to reason using only these for a downstream task. For example, if we consider a hidden layer with 800 components and U=16 competitors, it is easier to reason and investigate the 50 activated components instead of investigating the effect all components in conventional architectures.
>
> We believe that this argument clarifies our sparsity-interpretability claims and resolves the worries of the Reviewer. In the context of our future work, we aim to extensively analyse and evaluate how this data-driven sparsity mechanism can further facilitate interpretability of modern architectures while considering a variety of different ante and post-hoc methods starting from linear probe methods up to and further than Concept Bottleneck Models.
>
> >  The qualitative results provided in the paper are limited, particularly regarding the comparison between applying the WTA mechanism and not applying it.
>
> Considering the radically different mode of operation of Competitive Vision Networks and conventional architectures, it is not easy to draw a distinction between the methods on a qualitative basis since there isn't a one-to-one matching. This point is emphasized in the main text in line 305.
>
> There are instances where both the conventional networks and the CVNs do not provide the optimal results, e.g., in Fig. 3 Neuron 242; this result was purposely included in the Figure to highlight this fact. This is also case with the original CLIP-DISSECT work: therein, in Fig. 1 Neuron 683, some of the examples that highly activate the neuron have no relationship with the Neuron Identification label. In turn, this is why we considered the quantitative analysis in Table 2 using the metrics introduced in the CLIP-Dissect paper. Using the introduced metrics, it is  quite clear that using the LWTA mechanism yields a higher cosine similarity between the intermediate neuron descriptions and the ground truth labels compared to the conventional method that does not employ the LWTA mechanism; this illustrates the improvement in performance. Devising other quantitative methods for comparing conventional and LWTA-based Vision Networks, especially in the context of Network Dissection where one-to-one direct comparison is rendered difficult, is a substantial research challenge that we aim to nevertheless address in the future.
>
> > The dependence on the interpretability similarity metric raises questions about the generalizability of the results.
>
> The reviewer is concerned about the generalizability of the results.  However, as mentioned in the main text (and we will further expand in the camera ready), one of the main ``limitations'' of the proposed framework is the dependence on and the comparison restriction using only the CLIP-DISSECT approach. Concerning the visualization techniques issue that the reviewer raises, we believe this was addressed in the previous question. For the evaluation metrics of the results we considered both the best performing metric that the original publication uses (Soft WPMI), that substantially improved upon their other considered metrics, as well as the introduced Jensen Shannon divergence. We validated the improved performance using the cosine similarity that the original publication uses but apart from that, we believe that considering and introducing even more measures and techniques, goes beyond the rational scope of this work.

---

> > ### Comment · Reviewer_kXaB · 2023-08-16
> >
> > Thank you for the clarifications. After reading the rebuttal, I am more convinced that the assumption that sparser representations are more interpretable is not correct. We can not make such assumption, maybe neurons activate for few images but there is not a clear, human interpretable pattern in the set of image that helps humans interpret the activity. So, I would like to keep my score as this assumption is at the core of this work.

---

> > > ### Author Response · Authors · 2023-08-16
> > >
> > > We thank the reviewer for his response, allowing us to further add to this point.
> > >
> > > The underlying argument that sparsity is intuitively related to interpretability is also noted by reviewer CPpE "The argument that achieves higher sparsity and thus higher interpretability intuitively makes sense, but is this supported empirically?".
> > >
> > > Sparsity and Interpretability have recently gained a lot of attention with recent works aiming to create more interpretable layers via sparsification in the context of Concept Bottleneck Models in particular [1,2]. Even though these operate on the basis of a sparse linear final layer, the premise is the same: "A sparse linear model allows us to reason about the network’s decisions in terms of a significantly **smaller set of deep features**" [1].
> > >
> > > In conventional ReLU/GeLU-based vision networks, **ALL** neurons are activated for **ALL** examples, and neuron identification and intepretability are examined and analysed on this basis. In this context, we posit that activating only *a subset of neurons for each example* in the context of CVNs, facilitates the examination of the activated neurons (which are tied to particular concepts via Neuron Identification) and allows for reasoning on **that specific subset** in a similar manner to the seminal work of [1].
> > >
> > > [1] Eric Wong, Shibani Santurkar, and Aleksander Madry, Leveraging Sparse Linear Layers for Debuggable Deep Networks, ICML 2021
> > > [2] Tuomas Oikarinen, Subhro Das, Lam M. Nguyen, Tsui-Wei Weng, Label-Free Concept Bottleneck Models, ICLR 2023

---

> > > > ### Comment · Reviewer_kXaB · 2023-08-18
> > > >
> > > > Sparsity is not intuitively related to interpretability. You could have all neurons activated for all examples and have interpretable neurons, and also have neurons sparsely activated and have interpretable neurons too.
> > > >
> > > > Regarding your second comment: Could you please point me to the evidence that shows that ALL neurons are activated for ALL examples?

---

> > > > > ### Author Response · Authors · 2023-08-20
> > > > >
> > > > > Regarding the second comment, we point the reviewer to our previous answer:
> > > > >
> > > > > > We consider the official pretrained conventional DeiT-T checkpoint with the standard GeLU activation and report the sparsity level: this is obtained by averaging the number of activated neurons after the application of the GeLU nonlinearity in the MLP block of DeiT and considering each neuron as active if its absolute value is greater than a small constant, i.e.,
> > > > >  to account for rounding errors. For this test, we average over all the test examples of the ImageNet-1k validation set in all layers of DeiT. We observed that 98% of the neurons are active in this case. Using the exact same implementation for our LWTA-based DeiT-tiny implementation with 16 competitors, yields the expected 6.25% sparsity.
> > > > >
> > > > > Thus, we correct our phrasing that ALL neurons are active to **on average 98% of neurons are active** for the DeiT-T when using the ImageNet-1k validation set as a probing dataset, which we consider very substantial empirical evidence.
> > > > >
> > > > > > Sparsity is not intuitively related to interpretability. You could have all neurons activated for all examples and have interpretable neurons, and also have neurons sparsely activated and have interpretable neurons too.
> > > > >
> > > > > Please also see the response to reviewer BNcZ: We are not claiming theoretical or empirical guarantees on specialization and sparsification.
> > > > >
> > > > > However, we have pointed previous works that consider the sparsity effect on interpretability [1]. This seminal work considers a variety of settings, even using crowdsourcing to validate the results. Again, this does not mean that sparsity is the gold-standard, but sparsity surely facilitates the examination of the individual neurons, at least in the considered setting.
> > > > >
> > > > > Specifically, as noted in our response to reviewer BNcZ, in both conventional and LWTA-based networks neurons are tied to specific concepts through CLIP-Dissect and a probing dataset. Thus, after applying CLIP-Dissect each neuron has been assigned a specific functionality that remains fixed for inference. There are neurons tied to the color yellow, to aquaculture or to trains, to dashboard, e.t.c.
> > > > >
> > > > > When 98% neurons on average are active for each example when using ReLU/GeLU-based networks, this apparently yields conflicting information and burdens the examination of the decision process. As mentioned in the response to reviewer BNcZ, activating yellow, red, aquaculture, green and on average 98% of neurons that have substantially different functionality for each example surely does not yield an interpretable process.
> > > > >
> > > > > In other settings and other frameworks what the reviewer suggests can surely be valid. In the context of Network Dissection and conventional vision networks, the empirical evidence suggest that considering a different mode of operation such as the proposed LWTA networks may be imperative.
> > > > >
> > > > > [1] Eric Wong, Shibani Santurkar, and Aleksander Madry, Leveraging Sparse Linear Layers for Debuggable Deep Networks, ICML 2021

---

> > > > > ### Author Response · Authors · 2023-08-21
> > > > >
> > > > > Following up on the discussion with the reviewers, we have obtained an analysis on the activated neurons for a random example from the ImageNet-1k validation dataset, where we showcase the benefits of CVNs in this setting. Please see the response to reviewer CPpE.

---

> > > > > > ### Comment · Reviewer_kXaB · 2023-08-21
> > > > > >
> > > > > > The authors are not replying to my point. I downgraded my score.

---

> > > > > > > ### Author Response · Authors · 2023-08-21
> > > > > > >
> > > > > > > Can the reviewer highlight which exact point we did not reply to? We believe that we have addressed all issues raised by the reviewer in a thorough manner.

---

### Official Review · Reviewer_BNcZ · 2023-07-27

**Soundness:** 3 good
**Presentation:** 3 good
**Contribution:** 3 good
**Rating:** 5
**Confidence:** 2

**Summary:**

The work proposes a interpreatable network that uses concepts from competition networks specifically there are for a specific output multple copies of the neuron and the neurons compete to provide the winning activation for that output. The method proposes MLP and  convolutional versions of their network. The work also provides evidence of interpreatability by using CLIP-DISSECT and shows that their method is more interpreatable while using same number of parameters as DiET.

**Strengths:**

1. The method is quite simple and elegant in its motivation

2. Existing modules can be swapped with the proposed block without any changes

**Weaknesses:**

1. The method is essentially a stochastic version of maxpooling where the deterministic version is exactly maxpooling. Is there an ablation to show how the stochastic version is more appropriate than a deterministic one?

2. Having 2 competetors seems to provide the best performance whereas using more decreases the performance of the network. Is this because of using less "effective neurons" (as the model size kept same) or with more competition the network inherently starts degrading? If it's the second case than it does not put competition networks in a good light.

3. It seems wasteful that with more competition exponentially lower number of neurons in the model are activated.

5. Can it be used in networks for other tasks? Segmentation, object detection etc.?

**Questions:**

See the weakness.

1. Gumbel softmax generally requires a bit of finetuning. How stable is the gumbel softmax training on different datasets?

---

> ### Author Rebuttal · Authors · 2023-08-08
>
> We thank the reviewer for his insightful analysis and crucial comments and questions.
>
> > 1. The method is essentially a stochastic version of maxpooling where the deterministic version is exactly maxpooling. Is there an ablation to show how the stochastic version is more appropriate than a deterministic one?
>
> There is one pivotal difference between maxpooling and the LWTA mechanism: in the LWTA context we do not just output the max value but instead, and since we do not perform any reduction operation, we retain the positional information of both the active and inactive neurons, yielding *sparse* representations for each layer; this in turn allows for highly different patterns for each example, facilitating learning through specialization. See also the discussion in [1].
>
> We did not perform an ablation study to assess the relative performance between the deterministic and stochastic variants considering the significant computational complexity. However, several prior works (line 137, main text) have consistently exhibited that the stochastic LWTA formulation outperforms its deterministic counterpart in various settings. Moreover, only the stochastic variant allows for  introducing an additional metric for matching neurons to labels, i.e., the proposed Jensen-Shannon divergence.
>
> > 2. Having 2 competitors seems to provide the best performance whereas using more decreases the performance of the network. Is this because of using less "effective neurons" (as the model size kept same) or with more competition the network inherently starts degrading? If it's the second case than it does not put competition networks in a good light.
>
> We thank the reviewer for this question, as it will help clarifying some key points.  The goal of this work was not to achieve state-of-the-art performance for Vision tasks (main text, line 283), but instead provide empirical evidence towards a viable and radically different LWTA-based mode of operation for both Vision Networks and interpetability overall. To this end, we retained the training procedures for conventional ReLU/GeLU-based networks that have been rigorously optimized. We strongly believe that the observed performance difference is not a matter of using ``less effective'' neurons, but that that these procedures are highly sub-optimal for the mode of operation of LWTA; thus, extensive ablation studies and optimization will alleviate this issue (if one aims for classification accuracy as was the case with the baselines). However, this is not feasible in the scope of a single publication.
>
> Moreover, prior works in pruning and sparsity mechanisms have provided substantial empirical evidence of the significant redundancy of components in modern architectures. This fact further supports our claim that is not a problem of using less effective neurons but rather that CVNs have yet to reach their full potential. This work constitutes a first, and in our view very attentive, investigation of the behavior of CVNs on large scale.
>
> > 3. It seems wasteful that with more competition exponentially lower number of neurons in the model are activated.
>
> Similarly to the previous points, using a lower number of neurons may seem wasteful from a first glance but it also comes with significant benefits in various settings from meta learning to adversarial robustness (see also [1,2]). In the context of interpretability, activating, e.g., $1/16$ of the original neurons per each layer, potentially facilitates the process of neuron specialization: since not all neurons are activated for each example, neurons can specialize to only the features of the examples for which they are active. After Neuron Identification is performed, we can even narrow down the neurons that were activated by a particular test input and try to reason using only these for a downstream task. For example, if we consider a hidden layer with 800 components and $U=16$ competitors, it is easier to reason and investigate the 50 activated components instead of investigating the effect all components in conventional architectures. This can be applied in a variety of settings such as Concept Bottleneck Models and other approaches that we aim to explore in the future.
>
> > 4. Can it be used in networks for other tasks? Segmentation, object detection etc.?
>
> WTA-based activations can in principle be used in any kind of network replacing any of the conventional nonlinearities, in any kind of setting and task. For example, it has recently been explored in the context of end-to-end sign language translation [3]. Intepretability in the context of classification is a significant research and societal challenge, thus we focused on this setting. We hope that this work can facilitate the application of this underexplored but important mechanism to every ML task.
>
> > Q1: Gumbel softmax generally requires a bit of finetuning. How stable is the gumbel softmax training on different datasets?
>
> For training all our networks we set the temperature of the Gumbel-Softmax relaxation to a fixed $\tau = 0.67$ value as suggested in [4]. We found this choice to be extremely stable without requiring any additional finetuning. It is however possible that finetuning this value may provide further improvements accuracy-wise, especially when altering the number of competitors (essentially the number of categories of the distribution). We aim to explore training improvements in the near future.
>
>
> [1] Compete to Compute, R. Srivastava, J. Masci, S. Kazerounian, F. Gomez, J. Schmidhuber, NIPS, 2013
>
> [2] Konstantinos Kalais and Sotirios Chatzis. Stochastic deep networks with linear competing units for model-agnostic meta-learning, ICML, 2022
>
> [3]  Stochastic Transformer Networks with Linear Competing Units: Application to end-to-end SL Translation, Voskou et al., ICCV, 2021
>
> [4] Chris J. Maddison, Andriy Mnih, and Yee Whye Teh. The concrete distribution: A continuous relaxation of discrete random variables. In Proc. ICLR, 2017.

---

> > ### Comment · Reviewer_BNcZ · 2023-08-18
> > **Thank you**
> >
> > Thank you for extensive comments on my review. The review adresseses many of my concerns.
> >
> > > 1. The method is essentially a stochastic version of maxpooling where the deterministic version is exactly maxpooling. Is there an ablation to show how the stochastic version is more appropriate than a deterministic one?
> >
> > Answer is mostly convincing. Thank you for additionally pointing to citing [1]. Argument against is that reduction can be easily undone for maxpooling, (e.g. done in backpropagation).
> >
> > > 2. Having 2 competitors seems to provide the best performance whereas using more decreases the performance of the network. Is this because of using less "effective neurons" (as the model size kept same) or with more competition the network inherently starts degrading? If it's the second case than it does not put competition networks in a good light.
> >
> > Answer is mostly convincing. Although if current setup is suboptimal it puts the work at a disadvantage. The work is skipping a step in the discovery process. There should exist a well optimized LWTA based network to start with, before performing any kind of analysis on their advantages.
> >
> > > 3. It seems wasteful that with more competition exponentially lower number of neurons in the model are activated.
> >
> > Answer is mostly convincing. I agree with reviewer kXaB that lower number of neurons may not necessarily mean better interpreatability. There are still 50 neurons interacting with each other in the next step. There is no guarantee that each neuron is going to be specialized.
> >
> > > 4. Can it be used in networks for other tasks? Segmentation, object detection etc.
> >
> > The answer is satisfactory.
> >
> > > Q1: Gumbel softmax generally requires a bit of finetuning. How stable is the gumbel softmax training on different datasets?
> >
> > The answer is satisfactory.

---

> > > ### Author Response · Authors · 2023-08-20
> > >
> > > We thank the reviewer for the comments on our response.
> > >
> > > > Although if current setup is suboptimal it puts the work at a disadvantage.
> > >
> > > We agree with the reviewer that it would be optimal to consider the best-optimized LWTA network. However, this process requires a very significant amount of computational resources and computational time in general. Nevertheless, in this work, we trained LWTA-based networks that reach or perform better than their ReLU/GeLU based counterparts, in which many post-hoc interpretability methods have already been applied. This is the first time that something like this has been achieved and  we hope that this works further advances investigations of CVNs.
> > >
> > > > There is no guarantee that each neuron is going to be specialized.
> > >
> > > We agree with the reviewer on this point and we are not claiming or providing any theoretical or empirical guarantees about ideal specialization or sparsification. What we posit though is that the lower number of neurons to examine can indeed facilitate the investigation of the networks decision process.
> > >
> > > After training a specific backbone network, we apply the post-hoc CLIP-Dissect method. In this context, each neuron is tied to a specific concept according to how these neurons are activated when presented with the probe dataset. This is true for both the conventional and the LWTA-based networks.
> > >
> > > However, in conventional networks most neurons are active (see response to reviewer kXaB); specifically, we observed that on average, 98% on neurons are active for each example (averaged over the validation set of ImageNet-1k). Thus, even neurons that are tied to very different concepts are very likely active: the neuron tied to the color yellow, as neuron 50 in Fig. 3, is probably active for an image as the manatee that activates neuron 700 that is tied to aquaculture. This also applies to a potential activation of neuron 242 and neuron 46 from the same figure, *since on average 98% of neurons are active per example*.
> > >
> > > In contrast, and in the context of the considered framework, we argue that having the sparse mode of activation of LWTA can reduce the number of neurons that activate for each example (with 16 competitors, 6.25% active neurons); neurons that we can examine to which concepts they are tied to (since we have already applied clip-dissect) and try to investigate and infer the process on a substantially smaller subset of neurons.

---

> > > > ### Comment · Reviewer_BNcZ · 2023-08-20
> > > > **Thank you**
> > > >
> > > > Thank you for the clarification, I raise my score to boderline accept.

---

### Decision · Program_Chairs · 2023-09-21

**Decision:**

Accept (poster)

**Comment:**

This paper proposed a framework for creating interpretable vision networks based on stochastic winner-take-all layers. Key results show that the framework leads to a small and interpretable subset of neurons per data point.

Strengths
- The use of stochastic winner-take-all layers in the context of network dissection is novel and elegant.
- The use of LWTA does not impact the accuracy of the underlying model making this a useful technique for interpretability

Weaknesses
- Qualitative results in the paper can be improved. I suggest the authors add more such results in the appendix.
- Paper writing is confusing overall and should be simplified. RcrQf & RCppE made good suggestions on how to do this.

Reviews

This paper received mixed feedback from the reviewers. However, after thoroughly reading the reviewer responses and the author responses, the AC decided to accept the paper since the authors did provide satisfactory responses to each of the reviewers.

Justification

While the reviewers raised objections about the claims of sparsity in the representation, the paper does not suggest that this property should always hold. All the reviewers recognized the simplicity of the proposed approach. The AC also believes this work is more practical since it doesn’t trade off accuracy for intrepretability.
Overall, this paper proposes an interesting way to construct neural networks and should be valuable to the community.